# POWER CHARACTERIZATION OF NOISY QUANTUM KERNELS

## ABSTRACT

Quantum kernel methods have been widely recognized as one of promising quantum machine learning algorithms that have potential to achieve quantum advantages. In this paper, we theoretically characterize the power of noisy quantum kernels and demonstrate that under global depolarization noise, for different input data the predictions of the optimal hypothesis inferred by the noisy quantum kernel approximately concentrate towards some fixed value. In particular, we depict the convergence rate in terms of the strength of quantum noise, the size of training samples, the number of qubits, the number of layers affected by quantum noises, as well as the number of measurement shots. Our results show that noises may make quantum kernel methods to only have poor prediction capability, even when the generalization error is small. Thus, we provide a crucial warning to employ noisy quantum kernel methods for quantum computation and the theoretical results can also serve as guidelines when developing practical quantum kernel algorithms for achieving quantum advantages.

## 1 INTRODUCTION

### 1.1 BACKGROUND

A main objective of machine learning is to design efficient and robust computation methods to make accurate predictions for unseen data by using experiences, even for large-scale problems (Zhang et al., 2022; Ergun et al., 2022; Lyle et al., 2022; Mohri et al., 2018; Wright & Ma, 2022). Quantum machine learning (QML) aims to explore the representational and computational power of quantum models to offer advantages beyond what is possible using classical models (Dunjko & Briegel, 2018; Anschuetz, 2022; Kübler et al., 2021; Landman et al., 2023; Huang et al., 2021a; Wang et al., 2023a; Jerbi et al., 2021; Liu et al., 2021; Huang et al., 2021b). Among different types of QML modes (Wittek, 2014; Schuld et al., 2015; Biamonte et al., 2017; Zhang & Ni, 2020; Li & Deng, 2022; Guan et al., 2021), quantum kernel methods have attracted increasing attention and shown great potential for developing powerful new applications (Havlíček et al., 2019; Schuld & Killoran, 2019).

In machine learning, the prediction error can be decomposed into the sum of the training error and the generalization error, where the so-called generalization depicts the difference between the prediction error on new data and the training error. To make accurate predictions on unseen data, both of the training and generalization errors should be small (Zhang et al., 2017). In classical machine learning, it is often much easier to achieve small training errors than to guarantee good generalization. However, in QML the main obstacle is training and it is often challenging to achieve good trainability. For QML models based on quantum neural networks (QNNs), their landscapes often suffer from vanishing gradients known as barren plateaus (McClean et al., 2018; Haug et al., 2021; Ortiz Marrero et al., 2021) and/or the existence of exponentially many local minima (You & Wu, 2021; Anschuetz & Kiani, 2022), which make the training of QNNs extremely difficult. Under noiseless scenarios, quantum kernel methods do not suffer from these trainability issues and thus can naturally achieve smaller training errors as compared to QNNs. This is because for quantum kernel methods, due to the fundamental representation theorem (Schölkopf et al., 2002; Mohri et al., 2018), the optimal parameters minimizing the training error can always be found when the landscape of the cost function is convex (Havlíček et al., 2019; Schuld & Killoran, 2019; 2022; Jerbi et al., 2023). Quantum kernel methods are widely believed to be representative for achieving practical quantum

advantages. The prediction advantages over some classical models by employing quantum kernel methods have been demonstrated in Huang et al. (2021a); Liu et al. (2021).

Although quantum kernel methods have shown great potential for achieving quantum advantages, most of the existing results focus on ideal quantum settings without noise. Since noise may severely degrade the performance of quantum kernels, with the current noisy intermediate-scale quantum (NISQ) devices, a natural and crucial question is: *What is the power of noisy quantum kernel methods?* In this paper, we theoretically characterize the power of noisy quantum kernels and prove that for a given number of training samples, once the number of layers affected by noise exceeds some threshold, the prediction capability of noisy kernels is very poor. These limitations are quantitatively demonstrated by an upper bound on the expected distance between the predictions of the worst hypothesis without prediction capability and the optimal hypothesis for noisy quantum kernels. The results provide insights for understanding power and limitations of quantum kernels in the NISQ era and guidelines for developing competitive quantum kernel algorithms.

## 1.2 RELATED WORK

Limitations of optimization and variational quantum algorithms on noisy quantum devices were investigated in Stilck França & Garcia-Patron (2021); De Palma et al. (2023). Exponentially tighter bounds on limitations of quantum error mitigation has been given in Quek et al. (2022). Their adopted noise model is either local depolarizing noise or some non-unital noise, applied on each qubit. However, their $N$-fold layerwise noise channel is stronger than the global depolarizing model adopted for power characterization in this work. Here, $N$ denotes the number of qubits. This is because with the same noise rate, the probability of the quantum state remaining unchanged under our noise is exponentially larger than that in their models. In this work we focus on investigating the limitations of noisy quantum kernel methods.

It was demonstrated in Thanasilp et al. (2022) that values of quantum kernels over different input data can be exponentially concentrated towards some fixed value under the Pauli noise. Similar to the above local depolarizing noise applied to each qubit, the Pauli noise assumption is also stronger than our global depolarizing model. In addition, their noise-induced concentration bound does not take into account of the size of training samples. Thus, for a given size of training samples, their result cannot tell how many noisy layers will cause poor prediction capability for quantum kernel methods.

The power of noisy quantum kernel methods under global depolarizing noise and sampling error was investigated in Wang et al. (2021). Their main result is informative only for shallow quantum circuits. In addition, their main result is based on the key assumption of zero training error. Such an assumption places a strong constraint and may limit the applicability of their main results in dealing with noisy kernels. As we will demonstrate in this work, in the presence of noise, the training error may be large and dominate the prediction error, making noisy quantum kernel methods fail.

## 1.3 OUR CONTRIBUTIONS

In this work, we propose a new figure of merit to depict the power and limitations of quantum kernel methods, especially the impact of quantum depolarization noise on their prediction capability.

Our main contribution is providing a theoretical characterization of prediction concentration for different input data of the optimal hypothesis inferred by noisy quantum kernels. The concentration speed is clearly depicted in terms of the strength of depolarization noise $\tilde{p}$, the size of training samples, the number of qubits $N$, the number of layers affected by quantum noises, and the number of measurement shots. The results are summarized in Fig. 1, where the red regions represent the situations in which noisy quantum kernel methods fail, namely, the prediction capability is very poor. Especially, even with exponentially many training samples like $q^N$ ($q > 1$), noisy quantum kernels fail once the number of layers affected by noise exceeds $N \log_{(1-\tilde{p})^{-2}} q$.

We remark that our results hold for a wide range of quantum embedding schemes as we assume little on the form of quantum encoding circuits. Moreover, our upper bounds can be applied to quantum circuits with a large number of qubits and deep depth, not only limited to the current available shallow circuits. Thus, our results not only serve as a warning for shallow NISQ circuits, but also provide guidelines for future quantum computation. In addition, our results complement the research

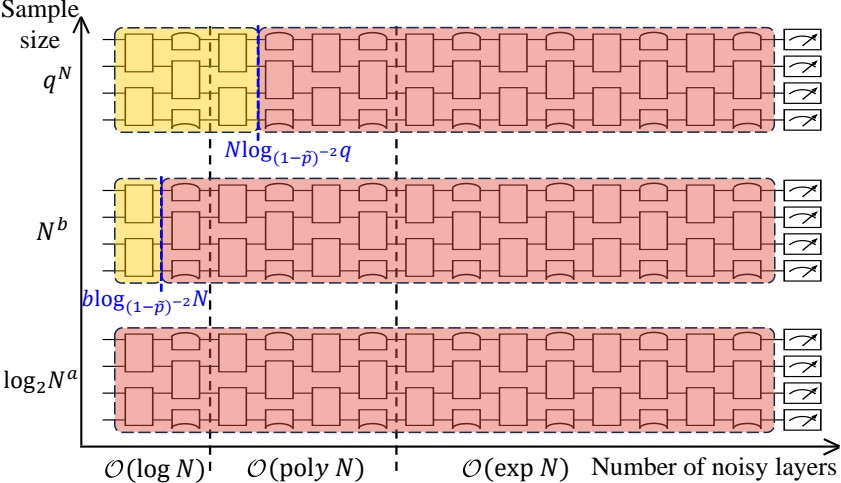

Figure 1: Summary of our main results. The red regions indicate the situations where noisy quantum kernel methods fail in prediction. Here, $N$ and $\tilde{p}$ denote the number of qubits and the strength of layerwise global depolarization noise, respectively. For logarithmically small training samples, noisy quantum kernel methods always fail. For training samples of polynomial size like $N^b$, noisy quantum kernel methods fail as long as the number of layers affected by noise exceeds $b \log_{(1-\tilde{p})^{-2}} N$. For training samples of exponential size like $q^N$ for some $q > 1$, noisy quantum kernel methods fail when the number of noisy layers exceeds $N \log_{(1-\tilde{p})^{-2}} q$.

on generalization of QML and indicate that a QML method having good generalization alone does not necessarily guarantee good prediction since the training error may be large, especially in the noisy cases. To achieve good prediction, both the training and generalization errors should be small.

This paper is organized as follows. In Section 2, we first introduce several preliminaries and then formulate the learning task with noisy quantum kernels. The main results are presented in Section 3. Numerical verifications are shown in Section 4. Section 5 concludes the paper.

## 2 Preliminaries and Framework

### 2.1 Kernel methods

Kernel methods are widely used in machine learning (Hofmann et al., 2008; Cho & Saul, 2009; Evgeniou et al., 2005; Shawe-Taylor & Cristianini, 2004). They are based on kernels or kernel functions, which implicitly define an inner product in a high-dimensional Hilbert space.

Assume that both training and test data are independent and identically distributed (i.i.d.) according to some fixed but unknown distribution $\mathcal{D}$ defined over $\mathcal{X} \times \mathcal{Y}$. Denote the training sample by $S = \{(\boldsymbol{x}_i, y_i)\}_{i=1}^n \subset \mathcal{X} \times \mathcal{Y}$. The kernel function $\mathcal{K}(\cdot, \cdot)$ is defined such that for $\boldsymbol{x}, \boldsymbol{x}' \in \mathcal{X}$,

$$\mathcal{K}(\boldsymbol{x}, \boldsymbol{x}') = \langle \Phi(\boldsymbol{x}), \Phi(\boldsymbol{x}') \rangle, \tag{1}$$

where $\Phi(\cdot)$ denotes a feature mapping that maps $\boldsymbol{x} \in \mathcal{X}$ to a high-dimensional Hilbert space called feature space with the inner product $\langle \cdot, \cdot \rangle$. A crucial benefit of kernel methods is that there is no need to explicitly define or compute the feature mapping $\Phi$. Instead, the performance of kernel-based learning depends on the kernel function $\mathcal{K}(\cdot, \cdot)$.

In kernel methods, the hypothesis function is typically chosen as

$$h(\boldsymbol{x}; \boldsymbol{\omega}) = \langle \boldsymbol{\omega}, \Phi(\boldsymbol{x}) \rangle, \tag{2}$$

where $\boldsymbol{\omega}$ is a vector in the feature space. Since the ultimate goal is to make accurate predictions for unseen data, the prediction error of a hypothesis $h(\boldsymbol{x}; \boldsymbol{\omega})$ with parameter $\boldsymbol{\omega}$ is taken to be the expected loss

$$R(\boldsymbol{\omega}) = \mathbb{E}_{(\boldsymbol{x},y) \sim \mathcal{D}} ([h(\boldsymbol{x}; \boldsymbol{\omega}) - y]^2). \tag{3}$$

As both the labels of unseen data and the distribution $\mathcal{D}$ are unknown, the prediction error is unavailable. The training error on the labeled sample $S$ is often taken as a proxy defined as

$$\widehat{R}_S(\boldsymbol{\omega}) = \frac{1}{n}\sum_{i=1}^{n}\left[h\left(\boldsymbol{x}_i;\boldsymbol{\omega}\right) - y_i\right]^2. \tag{4}$$

Notice that the prediction error can be decomposed as

$$R(\boldsymbol{\omega}) = \widehat{R}_S(\boldsymbol{\omega}) + \texttt{gen}(\boldsymbol{\omega}), \tag{5}$$

where $\texttt{gen}(\omega)$ is referred to as the generalization error. It is clear that to make accurate prediction, both of the training and generalization errors should be small.

To this end, we consider the following regularized optimization problem:

$$\min_{\boldsymbol{\omega}} \sum_{i=1}^{n}\left[h\left(\boldsymbol{x}_i;\boldsymbol{\omega}\right) - y_i\right]^2 + \lambda\langle\boldsymbol{\omega},\boldsymbol{\omega}\rangle, \tag{6}$$

where $\lambda>0$ is a hyperparameter. This convex minimization problem can be solved analytically, and the optimal parameter reads

$$\boldsymbol{\omega}^{\star} = \sum_{i,j=1}^{n}\Phi\left(\boldsymbol{x}_i\right)\left[\left(K + \lambda I\right)^{-1}\right]_{ij}y_j, \tag{7}$$

where the matrix $K \in \mathbb{R}^{n\times n}$, whose element $K_{ij} = \mathcal{K}\left(\boldsymbol{x}_i, \boldsymbol{x}_j\right) = \langle\Phi\left(\boldsymbol{x}_i\right), \Phi\left(\boldsymbol{x}_j\right)\rangle$.

## 2.2 QUANTUM KERNEL METHODS

In quantum computation, the carrier of information is qubits. For an $N$-qubit system, the quantum state can be mathematically represented as a positive semi-definite Hermitian matrix $\rho \in \mathbb{C}^{2^N \times 2^N}$ with $\text{Tr}(\rho) = 1$. Note that $\rho$ is called a pure state if $\text{rank}(\rho) = 1$, which can be represented in terms of a unit state vector $|\varphi\rangle$ as $\rho = |\varphi\rangle\langle\varphi|$, where $\langle\varphi| = |\varphi\rangle^{\dagger}$; otherwise, it is called a mixed state and can be decomposed as a convex combination of pure states.

For quantum computing, classical data $\boldsymbol{x} \in \mathcal{X}$ may be first embedded through an encoding quantum circuit (Havlíček et al., 2019; Schuld & Killoran, 2019; Lloyd et al., 2020; Schuld et al., 2021; Goto et al., 2021; Hubregtsen et al., 2022) denoted by $U_E(\cdot)$ as illustrated in Fig. 2(a). The encoded quantum state vector is

$$|\varphi\left(\boldsymbol{x}\right)\rangle = U_E\left(\boldsymbol{x}\right)|0\rangle^{\otimes N} \tag{8}$$

with the quantum state $\rho\left(\boldsymbol{x}\right) = |\varphi\left(\boldsymbol{x}\right)\rangle\langle\varphi\left(\boldsymbol{x}\right)|$ corresponding to a vector in the feature space with the inner product $\langle A, B\rangle = \text{Tr}\left[AB\right]$. The label $y$ of $\boldsymbol{x}$ can be generated through a quantum concept:

$$y = c\left(\boldsymbol{x}\right) = \text{Tr}\left[OU_{\text{QNN}}\rho\left(\boldsymbol{x}\right)U_{\text{QNN}}^{\dagger}\right], \tag{9}$$

where $O$ and $U_{\text{QNN}}$ represent the measurement operator and a specified quantum neural network, respectively. Without loss of generality, we assume that $\|O\|_2 \leq 1$. Here, $\|\cdot\|_2$ denotes the spectral norm, which is equal to the maximal singular value of the corresponding matrix.

In quantum kernel methods, we only need to employ a quantum circuit as illustrated in Fig. 2(b) to compute the quantum kernel functions as

$$\mathcal{K}\left(\boldsymbol{x}, \boldsymbol{x}'\right) = \text{Tr}\left[\rho\left(\boldsymbol{x}\right)\rho\left(\boldsymbol{x}'\right)\right] = \left|\langle\varphi\left(\boldsymbol{x}\right)|\varphi\left(\boldsymbol{x}'\right)\rangle\right|^2$$
$$= \text{Tr}\left[P_0 U_E^{\dagger}\left(\boldsymbol{x}'\right)U_E\left(\boldsymbol{x}\right)\left(|0\rangle\langle 0|\right)^{\otimes N}U_E^{\dagger}\left(\boldsymbol{x}\right)U_E\left(\boldsymbol{x}'\right)\right] \tag{10}$$

with the projector $P_0 = \left(|0\rangle\langle 0|\right)^{\otimes N}$. To demonstrate quantum advantages, the key is to construct a quantum encoding circuit $U_E(\cdot)$ such that patterns which are classically intractable can be recognized in the feature space (Liu et al., 2021).

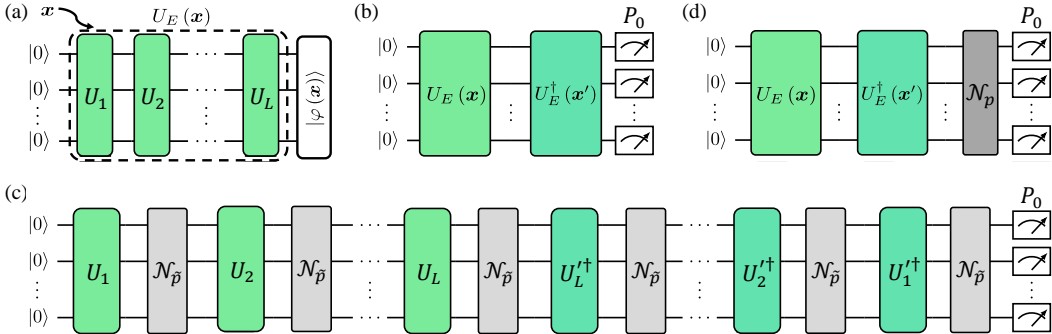

Figure 2: Quantum circuits employed in quantum kernel methods. (a) A general $L$-layer encoding circuit encodes the classical data $\boldsymbol{x}$ into the quantum state $|\varphi(\boldsymbol{x})\rangle$ through the quantum feature mapping $U_E(\cdot)$. (b) Quantum circuit utilized to compute quantum kernels in the ideal/noiseless case. The measurement operator is $P_0 = (|0\rangle\langle 0|)^{\otimes N}$. (c) After each layer of the encoding circuit, a global depolarizing channel with rate $\tilde{p}$ is applied. (d) The equivalent circuit of the noisy quantum circuit in (c). The total effect of all the quantum noise channels $\mathcal{N}_{\tilde{p}}$ can be effectively described by a global depolarizing channel $\mathcal{N}_p$ with $p = 1 - (1 - \tilde{p})^{2L}$.

Once the kernel matrix $K$ is obtained through Eq. (10), the remaining optimization is classical. For a given training sample $S = \{(\boldsymbol{x}_i, y_i)\}_{i=1}^{n}$, from Eqs. (2) and (7), the optimal hypothesis reads

$$h(\boldsymbol{x}) \triangleq h(\boldsymbol{x}; \boldsymbol{\omega}^\star) = \min\left\{1, \max\left\{-1, \operatorname{Tr}\left[\rho(\boldsymbol{x})\boldsymbol{\omega}^\star\right]\right\}\right\}$$

$$= \min\left\{1, \max\left\{-1, \sum_{i,j=1}^{n} \mathcal{K}(\boldsymbol{x}, \boldsymbol{x}_i)\left[(K + \lambda I)^{-1}\right]_{ij} y_j\right\}\right\}. \tag{11}$$

Here, the kernel function $\mathcal{K}(\boldsymbol{x}, \boldsymbol{x}_i)$ is also obtained via Eq. (10).

## 2.3 NOISY QUANTUM KERNELS

Up to now, we only consider the ideal setting, that is, the quantum circuits used to compute quantum kernels are unitary. However, in practice, particularly in the NISQ era, quantum circuits are susceptible to various quantum noises. In this work, we focus on the depolarization noise and consider its destructive impact on the prediction capability of quantum kernel methods. Our techniques may be generalized to other types of noise.

As illustrated in Fig. 2(c), when computing quantum kernels, as the noise model adopted in Wang et al. (2021), we assume that a global depolarizing channel with rate $\tilde{p}$ is applied after each layer of the ideal quantum circuit (illustrated in Fig. 2(b)), which reads

$$\mathcal{N}_{\tilde{p}}(\rho) = (1 - \tilde{p})\rho + \tilde{p}\frac{1}{D}I. \tag{12}$$

At first glance, our global depolarizing model is stronger than the so-called local noise models considered in Stilck França & Garcia-Patron (2021); De Palma et al. (2023); Thanasilp et al. (2022); Quek et al. (2022), which are in the form of $\mathcal{N}'_{\tilde{p}}(\rho) = \otimes_{j=1}^{N} \mathcal{N}'_j(\rho)$ with $\mathcal{N}'_j$ denoting either the single-qubit depolarizing noise, single-qubit Pauli noise, or single-qubit non-unital noise. In fact, our global model is weaker than these so-called local noise models for the problem in this paper. This is because at the same depolarizing rate $\tilde{p}$, the probability that the quantum state remains unchanged under our global noise is $(1 - \tilde{p})$, which is exponentially larger than that under the so-called local noise which is $(1 - \tilde{p})^N$. In addition, when presenting negative results concerning noisy quantum kernels, it is better to assume a relatively weaker noise model. Once the kernel methods fail under weaker noises, they fail under stronger ones in general. Note that the noise rate $\tilde{p} > 0$ in Eq. (12) can be arbitrarily small. Thus, it can depict the case where the noise influence is very weak, namely, after the noise the quantum state is left untouched with a very high probability $1 - \tilde{p}$.

It can be verified that the total effect of all the $2L$ quantum noise channels $\mathcal{N}_{\tilde{p}}$ can be effectively described by a global depolarizing channel

$$\mathcal{N}_p\left[\rho\left(\boldsymbol{x};\boldsymbol{x}'\right)\right]=(1-p)\,\rho\left(\boldsymbol{x};\boldsymbol{x}'\right)+p\frac{1}{D}I \tag{13}$$

applied after the whole ideal unitary channels. Here, $p=1-(1-\tilde{p})^{2L}$ with $L$ denoting the depth of the quantum encoding circuit $U_E\left(\cdot\right)$, $D=2^N$, and the ideal quantum output state $\rho\left(\boldsymbol{x};\boldsymbol{x}'\right)=|\varphi\left(\boldsymbol{x};\boldsymbol{x}'\right)\rangle\langle\varphi\left(\boldsymbol{x};\boldsymbol{x}'\right)|$, where $|\varphi\left(\boldsymbol{x};\boldsymbol{x}'\right)\rangle=U_E^\dagger\left(\boldsymbol{x}'\right)U_E\left(\boldsymbol{x}\right)|0\rangle^{\otimes N}$. The equivalent circuit is illustrated in Fig. 2(d) and the proof of the equivalence is given in Lemma A.5 in the Appendix. In fact, we can see that the exponent $L$ in the depolarization rate $p$ actually indicates the total number of layers affected by the depolarization noise $\mathcal{N}_{\tilde{p}}$ when implementing $U_E(\cdot)$.

Under the quantum depolarization noise, the noisy quantum kernel $\widetilde{\mathcal{K}}\left(\boldsymbol{x},\boldsymbol{x}'\right)$ reads

$$\widetilde{\mathcal{K}}\left(\boldsymbol{x},\boldsymbol{x}'\right)=\mathrm{Tr}\big\{P_0\,\mathcal{N}_p\left[\rho\left(\boldsymbol{x};\boldsymbol{x}'\right)\right]\big\}=(1-p)\,\mathcal{K}\left(\boldsymbol{x},\boldsymbol{x}'\right)+p\,\frac{1}{D}. \tag{14}$$

The corresponding noisy kernel matrix is $\widetilde{K}=(1-p)\,K+p\,\bar{K}$, with $\bar{K}=\frac{1}{D}J$, where $J$ denotes the matrix that has all entries 1. Accordingly, the optimal hypothesis in the presence of noise reads

$$\tilde{h}\left(\boldsymbol{x}\right)\triangleq\tilde{h}\left(\boldsymbol{x};\widetilde{\boldsymbol{\omega}}^\star\right)=\min\left\{1,\max\left\{-1,\sum_{i,j=1}^n\widetilde{\mathcal{K}}\left(\boldsymbol{x},\boldsymbol{x}_i\right)\left[\left(\widetilde{K}+\lambda I\right)^{-1}\right]_{ij}y_j\right\}\right\}. \tag{15}$$

Note that in the worst scenario where $p=1$, we have the noisy kernel denoted by $\bar{\mathcal{K}}$ with the property that for all $\boldsymbol{x},\boldsymbol{x}'\in\mathcal{X}$,

$$\bar{\mathcal{K}}\left(\boldsymbol{x},\boldsymbol{x}'\right)=\frac{1}{D}. \tag{16}$$

From Eq. (15), for new data $\boldsymbol{x}$, the corresponding optimal hypothesis returns the same value as

$$\bar{h}\left(\boldsymbol{x}\right)\triangleq\bar{h}\left(\boldsymbol{x};\bar{\boldsymbol{\omega}}^\star\right)=\frac{1}{D\lambda+n}\sum_{i=1}^n y_i, \tag{17}$$

which depends only on the training sample, not on the new data. Thus, it is completely uninformative for new data, and does not have any prediction capability at all.

In addition, when computing quantum kernels via quantum circuits, only a finite number of measurements are implemented in practice. Assume that $m$ measurements are implemented to compute each $\widetilde{\mathcal{K}}\left(\boldsymbol{x},\boldsymbol{x}'\right)$. Then the estimated noisy quantum kernels can be described as

$$\widehat{\mathcal{K}}\left(\boldsymbol{x},\boldsymbol{x}'\right)=\frac{1}{m}\sum_{k=1}^m V_k\left(\boldsymbol{x},\boldsymbol{x}'\right), \tag{18}$$

where each $V_k\left(\boldsymbol{x},\boldsymbol{x}'\right)$ is a Bernoulli random variable with the expectation being $\widetilde{\mathcal{K}}\left(\boldsymbol{x},\boldsymbol{x}'\right)$.

It can be verified that the random matrix $\widehat{K}+\lambda I$ is positive definite with probability of at least $1-ne^{-\lambda^2 m/4n}$ (see Lemma E.1). With the positive definiteness of $\widehat{K}+\lambda I$, the optimal hypothesis under the estimated noisy kernels reads

$$\hat{h}\left(\boldsymbol{x}\right)\triangleq\hat{h}\left(\boldsymbol{x};\widehat{\boldsymbol{\omega}}^\star\right)=\min\left\{1,\max\left\{-1,\sum_{i,j=1}^n\widehat{\mathcal{K}}\left(\boldsymbol{x},\boldsymbol{x}_i\right)\left[\left(\widehat{K}+\lambda I\right)^{-1}\right]_{ij}y_j\right\}\right\}. \tag{19}$$

## 3 MAIN RESULTS

In this section, we explicitly characterize the prediction capability of quantum kernel methods under quantum depolarization noise and measurement noise owing to finite shots. To this end, we consider a new figure of merit $\mathbb{E}_{(\boldsymbol{x},y)\sim\mathcal{D}}\left|\tilde{h}\left(\boldsymbol{x}\right)-\bar{h}\left(\boldsymbol{x}\right)\right|$. It describes the expected difference of the predictions

between the optimal hypothesis under the depolarization noise $\tilde{h}(\boldsymbol{x})$ and the worst hypothesis $\bar{h}(\boldsymbol{x})$, which essentially has no prediction capability at all.

In most of existing results, the performance of QML is usually evaluated by the upper bound of either the generalization error or the training error. The implicit assumption is that the learning algorithm can achieve small training error or generalization error, respectively, which does not always hold yet especially for NISQ settings. Now we present a result about the negative impact of noise on the power of quantum kernel methods.

**Theorem 3.1.** *For any $0<\delta<1$, with probability of at least $1-\delta$ over the draw of an i.i.d. sample $S = \{(\boldsymbol{x}_i, y_i)\}_{i=1}^n$ of size $n$, we have*

$$\mathbb{E}_{(\boldsymbol{x},y)\sim\mathcal{D}} \left| \tilde{h}(\boldsymbol{x}) - \bar{h}(\boldsymbol{x}) \right| \leq f\left( \frac{n}{\lambda}(1-p)\left(1+\frac{1}{D}\right) \right) + \frac{8\sqrt{Dn}}{D\lambda+n} + 6\sqrt{\frac{\log\frac{4}{\delta}}{2n}}, \qquad (20)$$

*where $f(z) = \frac{z+8\sqrt{\frac{z}{\lambda}}}{1-z}$ with $\lambda$ being the hyperparameter in Eq. (6), $p = 1-(1-\tilde{p})^{2L}$ is the depolarization rate with $L$ denoting the depth of $U_E(\cdot)$ in Eq. (8), and $D = 2^N$ is the dimension of the $N$-qubit state space.*

Theorem 3.1 holds for quantum circuits with arbitrary depth and width. Moreover, since we do not place strong constraints on the form of quantum encoding circuits $U_E$, our result holds for a wide class of encoding strategies. From Theorem 3.1, if the upper bound in Eq. (20) is small, then noisy quantum kernel methods fail in prediction for new data. Note that for the upper bound in Eq. (20), the first term $f(\cdot)$ converges to 0 if and only if its argument converges to 0, and the second term approaches to 0 as the increase of $D$ and $n$ (as long as $n$ is not in the order of $D = 2^N$).

To better characterize the limitations of noisy quantum kernel methods, we quantify the circuit depth $L$ and the size of training samples $n$ in terms of the number of qubits $N$. As illustrated by the vertical axis in Fig. 1, we consider three typical orders of the training size $n$. The red regions in Fig. 1 describe the ranges of the circuit depth $L$ such that the upper bound approaches to 0 as $N$ increases, making noisy quantum kernel methods fail. For example, noisy quantum kernel methods always fail for logarithmically small training samples, and even for training samples of exponential (polynomial) size $q^N$ with $q > 1$ ($N^b$ with $b > 1$), noisy quantum kernel methods fail as long as the number of noisy layers exceeds $N\log_{(1-\tilde{p})^{-2}} q$ ($b\log_{(1-\tilde{p})^{-2}} N$). In the yellow regions, our upper bound in Eq. (20) is uninformative, and needs to be further investigated. The demarcation lines of red and yellow regions are clearly depicted in Fig. 1.

To address practical and meaningful tasks, the scale of quantum circuits should be relatively large to generate a sufficient amount of expressibility. Thus, it is reasonable to utilize quantum circuits of polynomial depth. In this case, however, as illustrated in Fig. 1, even with exponentially large training samples, quantum kernel methods under noise may not predict well for unseen data. Therefore, our result provides a caveat for employing quantum kernel methods to demonstrate quantum advantages in the NISQ era.

The proof of Theorem 3.1 is mainly based on the following lemma.

**Lemma 3.2.** *Under the same setting as in Theorem 3.1, we have*

$$\mathbb{E}_{(\boldsymbol{x},y)\sim\mathcal{D}} \left| \tilde{h}(\boldsymbol{x}) - \bar{h}(\boldsymbol{x}) \right| \leq \lambda\|M_{\widetilde{K},\bar{K}}\|_2 + 8\sqrt{\left(1+\lambda\|M_{\widetilde{K},\bar{K}}\|_2\right)\|M_{\widetilde{K},\bar{K}}\|_2} + \frac{8\sqrt{Dn}}{D\lambda+n} + 6\sqrt{\frac{\log\frac{4}{\delta}}{2n}}, \tag{21}$$

*where $\|M_{\widetilde{K},\bar{K}}\|_2 = \left\| \left(\widetilde{K} + \lambda I\right)^{-1} - \left(\bar{K} + \lambda I\right)^{-1} \right\|_2$.*

To prove Theorem 3.1, we can further bound $\|M_{\widetilde{K},\bar{K}}\|_2$ as

$$\|M_{\widetilde{K},\bar{K}}\|_2 \leq \frac{\frac{n}{\lambda^2}(1-p)\left(1+\frac{1}{D}\right)}{1-\frac{n}{\lambda}(1-p)\left(1+\frac{1}{D}\right)}. \tag{22}$$

The detailed proof is given in Appendix C. We point out that it can be verified $\lambda\|M_{\widetilde{K},\bar{K}}\|_2$ bounds $\frac{1}{n}\sum_{i=1}^n \left| \tilde{h}(\boldsymbol{x}_i) - \bar{h}(\boldsymbol{x}_i) \right|$, which is the empirical difference between $\tilde{h}$ and $\bar{h}$.

In Theorem 3.1, we do not assume any prior information on the unknown distribution $\mathcal{D}$. In practice, to guarantee accurate predictions, learners prefer balanced training samples (Lindström et al., 2011; Wang et al., 2023b), where the amount of data belonging to different categories is the same.

**Definition 3.3.** (Balanced labels) In binary or multi-class classification tasks, assume that data are drawn from $\mathcal{X} \times \mathcal{Y}$ with respect to a discrete or continuous distribution $\mathcal{D}$. The labels $y$s generated from $\mathcal{D}$ are called balanced and normalized, if the labels for different categories are evenly distributed, and $\mathbb{E}_{(\boldsymbol{x},y)\sim\mathcal{D}} y = 0$.

With this additional prior information on the distribution $\mathcal{D}$, we can tighten the upper bound in Theorem 3.1 by reducing the second term as stated in the following corollary.

**Corollary 3.4.** *In addition to the setting stated in Theorem 3.1, assume that the labels $y$s generated from $\mathcal{D}$ are balanced. For any $0<\delta<1$, with probability of at least $1 - \delta$, we have*

$$\mathbb{E}_{(\boldsymbol{x},y)\sim\mathcal{D}} \left| \tilde{h}\left(\boldsymbol{x}\right) - \bar{h}\left(\boldsymbol{x}\right) \right| \leq f\left( \frac{n}{\lambda}\left(1-p\right)\left(1 + \frac{1}{D}\right) \right) + \frac{8\sqrt{2D\log\frac{4}{\delta}}}{D\lambda + n} + 6\sqrt{\frac{\log\frac{8}{\delta}}{2n}}. \tag{23}$$

The corresponding statements concerning the red and yellow regions and their boundaries in Fig. 1 still hold for the upper bound (23). Moreover, in this case, the worst hypothesis $\bar{h}$ behaves like a random-guess classifier and the hypothesis inferred by the noisy quantum kernel $\tilde{h}$ tends to perform no better than random guess in cases represented by the red regions.

We now consider the impact of the statistical measurement noise on the prediction capability of quantum kernel methods.

**Theorem 3.5.** *In addition to the setting stated in Theorem 3.1, assume that we perform $m$ measurements to compute the value of each kernel. For any $0<\delta<1$, with probability of at least $1 - \delta - ne^{-\lambda^2 m/4n}$, we have,*

$$\mathbb{E}_{(\boldsymbol{x},y)\sim\mathcal{D}} \left| \hat{h}\left(\boldsymbol{x}\right) - \bar{h}\left(\boldsymbol{x}\right) \right| \leq f\left( \frac{n}{\lambda}\left(1-p\right)\left(1 + \frac{1}{D}\right) + \frac{n}{\lambda}\sqrt{\frac{\log\frac{4n^2}{\delta}}{2m}} \right) + \frac{8\sqrt{Dn}}{D\lambda + n} + 6\sqrt{\frac{\log\frac{8}{\delta}}{2n}}, \tag{24}$$

*where $f\left(z\right) = \frac{z + 8\sqrt{\frac{z}{\lambda}}}{1-z}$.*

From Theorem 3.5, it is clear that once the number of measurement shots, $m = \Omega\left(n^{2+\epsilon}\right)$ with $\epsilon>0$, the upper bound Eq. (24) can be reduced to Eq. (20), which corresponds to the ideal case where an infinite number of measurement shots is implicitly assumed. This implies that when evaluating quantum kernels, the number of mesurement shots should be set at least $n^{2+\epsilon}$ to alleviate the negative impact of the measurement statistical noise on the prediction error.

## 4 NUMERICAL EXPERIMENTS

In this section, we validate the theoretical limitation of noisy quantum kernel methods via classification tasks. Following the results in Huang et al. (2021a) and Havlíček et al. (2019), which describe the power of quantum data and quantum kernel methods, respectively, we conduct experiments on the fashion-MNIST dataset (Xiao et al., 2017), which is more challenging for classification than on the MNIST data. For binary classification, we identify images as shirts or dresses.

As in Huang et al. (2021a), we first transform each original $28 \times 28$ grayscale image into a 10-dimensional vector using principal component analysis (Jolliffe, 2002). Then we use the IQP-type embedding circuit composed of single-qubit and 2-qubit unitary gates (Havlíček et al., 2019) to embed the 10-dimensional vector into the Hilbert space of $N = 10$ qubits. Specifically, the encoded quantum state vector reads

$$\left|\varphi\left(\boldsymbol{x}\right)\right\rangle = U_E\left(\boldsymbol{x}\right)\left|0\right\rangle^{\otimes N} = U_Z\left(\boldsymbol{x}\right)H^{\otimes N}U_Z\left(\boldsymbol{x}\right)H^{\otimes N}\left|0\right\rangle^{\otimes N}, \tag{25}$$

where $H^{\otimes N}$ denotes the Hadamard gate acting on all qubits in parallel, and

$$U_Z\left(\boldsymbol{x}\right) = \exp\left( \sum_{i=1}^{N} x_i Z_i + \sum_{i=1}^{N}\sum_{j=1}^{N} x_i x_j Z_i Z_j \right), \tag{26}$$

with $x_i$ denoting the $i$-th element of the vector $\boldsymbol{x}$ and $Z_i$ denotes the Pauli-$Z$ operator acting on the $i$-th qubit. Havlíček et al. (2019) stated that the embedding circuit $U_E(\boldsymbol{x})$ provides a quantum advantage as it is hard to simulate the circuit classically.

For binary classification, we employ the sign of the optimal hypothesis $\tilde{h}$ to predict labels of test samples, and take the frequency of misclassification over the training sample (test sample) as the proxy for the training error (prediction error). In the numerical experiments, we utilize both training and test samples of size $n = 500$, which is exponentially large with $N = 10$ ($n = q^N$ with $q = 1.86$). We set the strength of depolarization noise $\tilde{p} = 0.1$, and the regularization hyperparameter $\lambda = 0.5$.

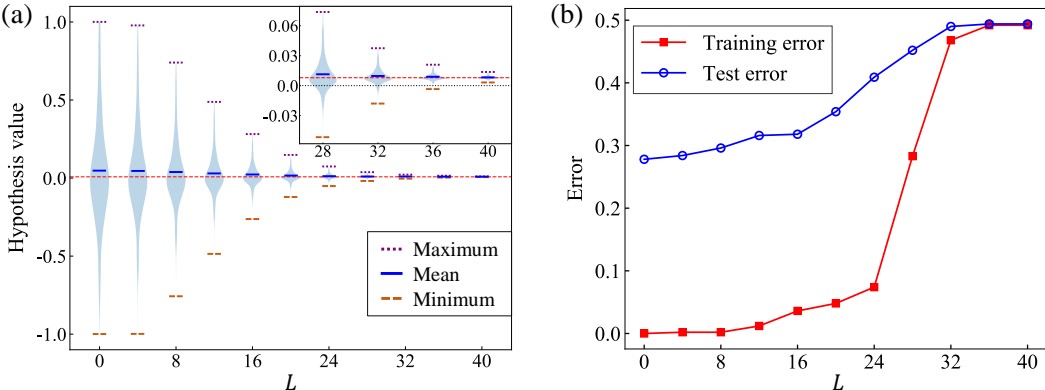

Figure 3: Prediction capability of quantum kernel methods under depolarization noise of different numbers of layers. Here, training sample size: $n = 500$, test sample size: $n = 500$, depolarization noise rate: $\tilde{p} = 0.1$, hyperparameter: $\lambda = 0.5$, and number of noisy layers: $L$. (a) Concentration of the noisy optimal hypothesis $\tilde{h}$ towards the worst hypothesis $\bar{h}$ as $L$ increases. The red dashed baseline denotes the value of $\bar{h}$, and the shaded areas denote the relative frequency of each hypothesis value over the 500 test samples, with the parma dotted, blue line, and dark orange dash represent the maximum, mean, and minimum values of $\tilde{h}$, respectively. (b) The training error (red square) and test error (blue circle) versus $L$. There is a phase transition for the training error at $L = 24$.

From Fig. 3(a), as the number of noisy layers increases, the values of $\tilde{h}$ on the test samples do converge to an uninformative value returned by the worst hypothesis $\bar{h}$. The convergency behavior coincides with the demarcation line as illustrated in Fig. 1, namely, when $L > \log_{(1-\tilde{p})^{-2}} 500 \approx 30$, $\mathbb{E}\left|\tilde{h}(\boldsymbol{x}) - \bar{h}(\boldsymbol{x})\right| \approx 0$. When $L = 40$, all the values of $\tilde{h}$ is positive, and all test samples will be labeled in the same class as that returned by $\bar{h}$, which is completely uninformative. Fig. 3(b) depicts the practical performances of the training error and test error as $L$ increases. For the training error, there is a phase transition at $L = 24$, which is owing to the accumulated noise in the circuit. The training error converges to the error of $\bar{h}$, and the test error tends to be independent of the test samples, and the prediction is no better than the random guess.

## 5 CONCLUSION

In this work, we investigate the power and limitations of quantum kernel methods under quantum global depolarization noise. We theoretically depict the concentration speed of predictions of the optimal hypothesis inferred by noisy quantum kernels. Our techniques can be generalized to investigate the impact of other typical quantum noises. Our results hold for a wide class of quantum encoding strategies, and are applicable not only on shallow NISQ circuits, but also on future large-scale quantum devices. Therefore, our results on the one hand make a clear warning against utilizing quantum kernel methods to demonstrate quantum advantages in the NISQ era, and on the other hand provide crucial guidelines in developing practical machine learning approaches for future quantum computation.

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

## A  TECHNICAL LEMMAS

For self-consistency, we firstly present two widely used concentration inequalities for independent random variables and matrices, respectively.

**Lemma A.1. (Hoeffding's inequality)** (Lemma D.1, Mohri et al. (2018)) *Let $X_1, \cdots, X_n$ be independent random variables with $X_i$ taking values in $[a_i, b_i]$ for all $i \in [n]$. Then, for any $\epsilon > 0$ and $S_n = \sum_{i=1}^n X_i$,*

$$\mathbb{P}\left[S_n - \mathbb{E}S_n \geq \epsilon\right] \leq e^{-2\epsilon^2 / \sum_{i=1}^n (b_i - a_i)^2},$$

$$\mathbb{P}\left[S_n - \mathbb{E}S_n \leq -\epsilon\right] \leq e^{-2\epsilon^2 / \sum_{i=1}^n (b_i - a_i)^2}.$$

**Lemma A.2. (Matrix Hoeffding)** (Corollary 4.2, Mackey et al. (2014)) *Let $Y^{(1)}, \cdots, Y^{(m)}$ be independent random Hermitian $n \times n$ matrices and $A^{(1)}, \cdots, A^{(m)}$ be deterministic Hermitian $n \times n$ matrices. Assume that for each $k \in [m]$,*

$$\mathbb{E}\left[Y^{(k)}\right] = 0 \quad \text{and} \quad \left[Y^{(k)}\right]^2 \preceq \left[A^{(k)}\right]^2.$$

*Here, $X \preceq Y$ means that the matrix $Y - X$ is positive semi-definite. Then, for all $t \geq 0$,*

$$\mathbb{P}\left[\lambda_{\max}\left(\sum_{k=1}^m Y^{(k)}\right) \geq t\right] \leq n e^{-t^2/2\sigma^2}, \tag{27}$$

$$\mathbb{P}\left[\lambda_{\min}\left(\sum_{k=1}^m Y^{(k)}\right) \leq -t\right] \leq n e^{-t^2/2\sigma^2}, \tag{28}$$

where $\sigma^2 = \frac{1}{2}\left\|\sum_{k=1}^{m}\left\{\left[A^{(k)}\right]^2 + \mathbb{E}\left[Y^{(k)}\right]^2\right\}\right\|_2$, $\lambda_{\max}(A)$ and $\lambda_{\min}(A)$ denote the maximal eigenvalue and the minimal eigenvalue of matrix $A$, respectively.

We can directly obtain the following equivalent form of Eq. (28):

$$\mathbb{P}\left[\lambda_{\min}\left(\sum_{k=1}^{m} Y^{(k)} + tI\right) \geq 0\right] \geq 1 - ne^{-t^2/2\sigma^2}. \tag{29}$$

That is, under the same assumptions as in Lemma A.2, for all $t \geq 0$, with probability of at least $1 - ne^{-t^2/2\sigma^2}$, the random matrix $\sum_{k=1}^{m} Y^{(k)} + tI$ is positive semi-definite.

Secondly, we present a basic result in machine learning, which provides an upper bound on the generalization error.

**Lemma A.3.** (Theorem 3.3, Mohri et al. (2018)) *Let $\mathcal{G}$ be a family of functions mapping from $\mathcal{Z}$ to $[0, 1]$. Then, for any $0<\delta<1$, with probability of at least $1 - \delta$ over the draw of an i.i.d. sample $S = (z_1, \ldots, z_n)$ of size $n$ with elements in $\mathcal{Z}$, the following inequality holds for all $g \in \mathcal{G}$:*

$$\mathbb{E}_{z}\left[g\left(z\right)\right] \leq \frac{1}{n}\sum_{i=1}^{n} g\left(z_i\right) + \frac{2}{n}\mathbb{E}_{\boldsymbol{\sigma}}\left[\sup_{g\in\mathcal{G}}\sum_{i=1}^{n} \sigma_i g\left(z_i\right)\right] + 3\sqrt{\frac{\log\frac{2}{\delta}}{2n}},$$

*where $\boldsymbol{\sigma} = (\sigma_1, \cdots, \sigma_n)^{\mathrm{T}}$ with $\sigma_i$s independent uniform random variables taking value in $\{-1, +1\}$.*

Here, $\hat{\mathcal{R}}_S(\mathcal{G}) = \mathbb{E}_{\boldsymbol{\sigma}}\left[\sup_{g\in\mathcal{G}}\frac{1}{n}\sum_{i=1}^{n} \sigma_i g\left(z_i\right)\right]$ is referred to as the empirical Rademacher complexity of the set $\mathcal{G}$ with respect to the sample $S$.

Thirdly, we present a lemma that relates the empirical Rademacher complexity of a new set of composite functions of a hypothesis in $\mathcal{H}$ and a Lipschitz function to the empirical Rademacher complexity of the hypothesis set $\mathcal{H}$.

**Lemma A.4. (Talagrand's lemma)** (Lemma 5.7, Mohri et al. (2018)) *Let $\Phi_1, \cdots, \Phi_n$ be $l$-Lipschitz functions from $\mathbb{R}$ to $\mathbb{R}$ and $\boldsymbol{\sigma} = (\sigma_1, \cdots, \sigma_n)^{\mathrm{T}}$ whose elements are independent uniform random variables taking value in $\{-1, +1\}$. Then, for any hypothesis set $\mathcal{H}$ of real-valued functions, the following inequality holds:*

$$\frac{1}{n}\mathbb{E}_{\boldsymbol{\sigma}}\left[\sup_{h\in\mathcal{H}}\sum_{i=1}^{n} \sigma_i \left(\Phi_i \circ h\right)\left(x_i\right)\right] \leq \frac{l}{n}\mathbb{E}_{\boldsymbol{\sigma}}\left[\sup_{h\in\mathcal{H}}\sum_{i=1}^{n} \sigma_i h\left(x_i\right)\right].$$

At last, we present the lemma concerning the total effect of all the quantum depolarizing channels $\mathcal{N}_{\tilde{p}}$.

**Lemma A.5.** (Lemma 2, Wang et al. (2021)) *For an $L$-layer quantum circuit $U = \prod_{l=1}^{L} U_l$ or channel $\mathcal{E} = \mathcal{E}_L \circ \cdots \circ \mathcal{E}_1$, the noise model where a global depolarizing channel $\mathcal{N}_{\tilde{p}}$ acts after each (unitary or completely positive trace preserving) layer is equivalent to a global depolarizing channel $\mathcal{N}_p$ following the entire quantum circuit or channel, where $p = 1 - (1 - \tilde{p})^L$. That is,*

$$\mathcal{N}_{\tilde{p}}\left\{U_L \cdots \mathcal{N}_{\tilde{p}}\left[U_2\mathcal{N}_{\tilde{p}}\left(U_1\rho U_1^{\dagger}\right)U_2^{\dagger}\right]\cdots U_L^{\dagger}\right\} = \mathcal{N}_p\left(U\rho U^{\dagger}\right),$$

$$\mathcal{N}_{\tilde{p}} \circ \mathcal{E}_L\left\{\cdots\mathcal{N}_{\tilde{p}} \circ \mathcal{E}_2\left[\mathcal{N}_{\tilde{p}} \circ \mathcal{E}_1\left(\rho\right)\right]\right\} = \mathcal{N}_p \circ \mathcal{E}\left(\rho\right).$$

# B    PROOF OF LEMMA 3.2

Firstly, we introduce the following lemma.

**Lemma B.1.** *Consider an optimal hypothesis function $h\left(\boldsymbol{x}; \boldsymbol{\omega}^{\star}\right)$ in the form of Eq. (11) associated with a specific kernel matrix $K_h$. For any $0<\delta<1$, with probability of at least $1 - \delta$ over the draw of an i.i.d. sample $S = \{(\boldsymbol{x}_i, y_i)\}_{i=1}^{n}$ of size $n$, the expected difference of the predictions between*

$h\left(\boldsymbol{x};\boldsymbol{\omega}^{\star}\right)$ and $\bar{h}\left(\boldsymbol{x}\right)$ is upper bounded as

$$
\underset{(\boldsymbol{x},y)\sim\mathcal{D}}{\mathbb{E}}\left|h\left(\boldsymbol{x};\boldsymbol{\omega}^{\star}\right)-\bar{h}\left(\boldsymbol{x}\right)\right| \leq \frac{1}{n}\left\|K_h(K_h+\lambda I)^{-1}Y-\frac{1}{D\lambda+n}JY\right\|_1
$$
$$
+\frac{8}{\sqrt{n}}\left\lceil\sqrt{Y^{\mathrm{T}}(K_h+\lambda I)^{-1}K_h(K_h+\lambda I)^{-1}Y}\right\rceil+6\sqrt{\frac{\log\frac{4}{\delta}}{2n}},
$$
$$
\tag{30}
$$

where $Y=(y_1,\cdots,y_n)^{\mathrm{T}}$, the vector norm $\|\cdot\|_p$ denotes the $l_p$-norm, and $\lceil\cdot\rceil$ represents the roundup function. Here, the first term in the right-hand side of Eq. (30) bounds the empirical difference $\frac{1}{n}\sum_{i=1}^{n}\left|h\left(\boldsymbol{x}_i;\boldsymbol{\omega}^{\star}\right)-\bar{h}\left(\boldsymbol{x}_i\right)\right|$.

*Proof.* The expected difference can be decomposed into the sum of the empirical difference and the so-called generalization as

$$
\underset{(\boldsymbol{x},y)\sim\mathcal{D}}{\mathbb{E}}\left|h\left(\boldsymbol{x};\boldsymbol{\omega}^{\star}\right)-\bar{h}\left(\boldsymbol{x}\right)\right| =\frac{1}{n}\sum_{k=1}^{n}\left|h\left(\boldsymbol{x}_k;\boldsymbol{\omega}^{\star}\right)-\bar{h}\left(\boldsymbol{x}_k\right)\right|
$$
$$
+\underset{(\boldsymbol{x},y)\sim\mathcal{D}}{\mathbb{E}}\left|h\left(\boldsymbol{x};\boldsymbol{\omega}^{\star}\right)-\bar{h}\left(\boldsymbol{x}\right)\right|-\frac{1}{n}\sum_{k=1}^{n}\left|h\left(\boldsymbol{x}_k;\boldsymbol{\omega}^{\star}\right)-\bar{h}\left(\boldsymbol{x}_k\right)\right|.
$$
$$
\tag{31}
$$

Firstly, we bound the empirical difference. According to the optimal hypothesis given in Eq. (11), for each data $\boldsymbol{x}_k$ in $S$,

$$
h\left(\boldsymbol{x}_k;\boldsymbol{\omega}^{\star}\right)=\min\left\{1,\max\left\{-1,\sum_{i,j=1}^{n}(K_h)_{ki}\left[(K_h+\lambda I)^{-1}\right]_{ij}y_j\right\}\right\}
$$
$$
=\min\left\{1,\max\left\{-1,\left[K_h\left(K_h+\lambda I\right)^{-1}Y\right]_k\right\}\right\}.
$$

Particularly, if $K_h=\bar{K}=\frac{1}{D}J$ which corresponds to the noisy kernel (16) in the worst scenario, then the optimal hypothesis $\bar{h}$ returns the same value for each $\boldsymbol{x}_k$ as

$$
\bar{h}\left(\boldsymbol{x}_k\right)=\frac{1}{D\lambda+n}\sum_{i=1}^{n}y_i=\left[\bar{K}\left(\bar{K}+\lambda I\right)^{-1}Y\right]_k=\left(\frac{1}{D\lambda+n}JY\right)_k.
$$

Thus, the empirical difference can be upper bounded as

$$
\frac{1}{n}\sum_{k=1}^{n}\left|h\left(\boldsymbol{x}_k;\boldsymbol{\omega}^{\star}\right)-\bar{h}\left(\boldsymbol{x}_k\right)\right| \leq \frac{1}{n}\sum_{k=1}^{n}\left|\left[K_h\left(K_h+\lambda I\right)^{-1}Y\right]_k-\left(\frac{1}{D\lambda+n}JY\right)_k\right|
$$
$$
=\frac{1}{n}\left\|K_h(K_h+\lambda I)^{-1}Y-\frac{1}{D\lambda+n}JY\right\|_1.
$$
$$
\tag{32}
$$

Next, we derive the upper bound of the generalization. Denote $d_{\boldsymbol{\omega}}\left(\boldsymbol{x}\right)=\left|h\left(\boldsymbol{x};\boldsymbol{\omega}\right)-\bar{h}\left(\boldsymbol{x}\right)\right|$. Since $d_{\boldsymbol{\omega}}\left(\boldsymbol{x}\right)\in[0,2]$, to utilize Lemma A.3, let $\mathcal{G}_{\gamma}=\left\{\frac{d_{\boldsymbol{\omega}}}{2}:\|\boldsymbol{\omega}\|\leq\gamma\right\}$, for $\gamma=1,\ 2,\ 3,\cdots$, where $\|\cdot\|$ denotes the Frobenius norm unless otherwise stated, and the subscript F has been omitted for brevity. Then from Lemma A.3, for any $\delta>0$ and $\gamma$, with probability of at least $1-\frac{\delta}{2\gamma^2}$ over the draw of an i.i.d. sample $S=\{(\boldsymbol{x}_i,y_i)\}_{i=1}^{n}$ of size $n$, the following inequality holds for any $\boldsymbol{\omega}$ with $\|\boldsymbol{\omega}\|\leq\gamma$:

$$
\underset{(\boldsymbol{x},y)\sim\mathcal{D}}{\mathbb{E}}\left[d_{\boldsymbol{\omega}}\left(\boldsymbol{x}\right)\right]-\frac{1}{n}\sum_{k=1}^{n}d_{\boldsymbol{\omega}}\left(\boldsymbol{x}_k\right)\leq\frac{2}{n}\underset{\boldsymbol{\sigma}}{\mathbb{E}}\left[\sup_{\|\boldsymbol{v}\|\leq\gamma}\sum_{k=1}^{n}\sigma_k d_{\boldsymbol{v}}\left(\boldsymbol{x}_k\right)\right]+6\sqrt{\frac{1}{2n}\log\frac{4\gamma^2}{\delta}}.
$$
$$
\tag{33}
$$

Thus, with probability of at least $1 - \sum_{\gamma=1}^{\infty} \frac{\delta}{2\gamma^2} \geq 1 - \delta$, the inequality (33) holds for all $\gamma$. Then for any $\boldsymbol{\omega} \in \mathcal{H}$, with probability of at least $1 - \delta$, we have

$$\mathop{\mathbb{E}}_{(\boldsymbol{x},y)\sim\mathcal{D}} [d_{\boldsymbol{\omega}}(\boldsymbol{x})] - \frac{1}{n}\sum_{k=1}^{n} d_{\boldsymbol{\omega}}(\boldsymbol{x}_k) \leq \frac{2}{n} \mathop{\mathbb{E}}_{\boldsymbol{\sigma}} \left[ \sup_{\|\boldsymbol{v}\| \leq \lceil\|\boldsymbol{\omega}\|\rceil} \sum_{k=1}^{n} \sigma_k d_{\boldsymbol{v}}(\boldsymbol{x}_k) \right] + 6\sqrt{\frac{1}{2n}\log\frac{4\lceil\|\boldsymbol{\omega}\|\rceil^2}{\delta}}. \tag{34}$$

Note that

$$d_{\boldsymbol{v}}(\boldsymbol{x}) = \left| h(\boldsymbol{x};\boldsymbol{v}) - \bar{h}(\boldsymbol{x}) \right| = \left| \min\left\{1, \max\left\{-1, \mathrm{Tr}\left[\rho(\boldsymbol{x})\boldsymbol{v}\right]\right\}\right\} - \frac{1}{D\lambda+n}\sum_{i=1}^{n} y_i \right|,$$

and the function $\Gamma(\cdot) = \left| \min\left\{1, \max\left\{-1, \cdot\right\}\right\} - \frac{1}{D\lambda+n}\sum_{i=1}^{n} y_i \right|$ is 1-Lipschitz. According to Lemma A.4, we have

$$\mathop{\mathbb{E}}_{\boldsymbol{\sigma}} \left[ \sup_{\|\boldsymbol{v}\| \leq \lceil\|\boldsymbol{\omega}\|\rceil} \sum_{k=1}^{n} \sigma_k d_{\boldsymbol{v}}(\boldsymbol{x}_k) \right] \leq \mathop{\mathbb{E}}_{\boldsymbol{\sigma}} \left[ \sup_{\|\boldsymbol{v}\| \leq \lceil\|\boldsymbol{\omega}\|\rceil} \sum_{k=1}^{n} \sigma_k \mathrm{Tr}\left[\rho(\boldsymbol{x}_k)\boldsymbol{v}\right] \right]$$

$$\leq \mathop{\mathbb{E}}_{\boldsymbol{\sigma}} \left[ \sup_{\|\boldsymbol{v}\| \leq \lceil\|\boldsymbol{\omega}\|\rceil} \|\boldsymbol{v}\| \left\| \sum_{k=1}^{n} \sigma_k \rho(\boldsymbol{x}_k) \right\| \right] \tag{35}$$

$$\leq \lceil\|\boldsymbol{\omega}\|\rceil \mathop{\mathbb{E}}_{\boldsymbol{\sigma}} \left[ \left\| \sum_{k=1}^{n} \sigma_k \rho(\boldsymbol{x}_k) \right\| \right]$$

$$\leq \lceil\|\boldsymbol{\omega}\|\rceil \sqrt{\mathop{\mathbb{E}}_{\boldsymbol{\sigma}} \left[ \sum_{i=1}^{n}\sum_{k=1}^{n} \sigma_i\sigma_k \mathrm{Tr}\left[\rho(\boldsymbol{x}_i)\rho(\boldsymbol{x}_k)\right] \right]} \tag{36}$$

$$\leq \lceil\|\boldsymbol{\omega}\|\rceil \sqrt{\mathrm{Tr}(K_h)} \tag{37}$$

$$\leq \lceil\|\boldsymbol{\omega}\|\rceil \sqrt{n}, \tag{38}$$

where Cauchy-Schwartz inequality and Jensen's inequality are applied to yield Eq. (35) and Eq. (36), respectively. To derive Eq. (37), we use the fact that $\sigma_1, \cdots, \sigma_n$ are independent uniform random variables taking value in $\{-1, +1\}$.

Combining Eqs. (34) and (38) with the following inequality

$$\sqrt{\log\frac{4\lceil\|\boldsymbol{\omega}\|\rceil^2}{\delta}} = \sqrt{\log\lceil\|\boldsymbol{\omega}\|\rceil^2 + \log\frac{4}{\delta}} \leq \sqrt{\log\lceil\|\boldsymbol{\omega}\|\rceil^2} + \sqrt{\log\frac{4}{\delta}} \leq \lceil\|\boldsymbol{\omega}\|\rceil + \sqrt{\log\frac{4}{\delta}},$$

it yields

$$\mathop{\mathbb{E}}_{(\boldsymbol{x},y)\sim\mathcal{D}} [d_{\boldsymbol{\omega}}(\boldsymbol{x})] - \frac{1}{n}\sum_{k=1}^{n} d_{\boldsymbol{\omega}}(\boldsymbol{x}_k) \leq \frac{8}{\sqrt{n}}\lceil\|\boldsymbol{\omega}\|\rceil + 6\sqrt{\frac{1}{2n}\log\frac{4}{\delta}}. \tag{39}$$

This holds for $\boldsymbol{\omega} = \boldsymbol{\omega}^{\star}$ which is in the form of Eq. (7) and satisfies

$$\|\boldsymbol{\omega}^{\star}\| = \sqrt{Y^{\mathrm{T}}(K_h + \lambda I)^{-1} K_h (K_h + \lambda I)^{-1} Y}. \tag{40}$$

Thus, by plugging Eqs. (32), (39), and (40) into Eq. (31), we prove Lemma B.1. $\square$

Secondly, we introduce the $L_2$-geometric difference $\|M_{K_h,\bar{K}}\|_2$ between a specific kernel matrix $K_h$ and the kernel matrix $\bar{K}$ as

$$\|M_{K_h,\bar{K}}\|_2 = \left\| (K_h + \lambda I)^{-1} - \left(\bar{K} + \lambda I\right)^{-1} \right\|_2, \tag{41}$$

and further bounds the right-hand side of Eq. (30).

**Lemma B.2.** *Under the same setting as in Lemma B.1, we have*

$$\mathop{\mathbb{E}}_{(\boldsymbol{x},y)\sim\mathcal{D}}\left|h\left(\boldsymbol{x};\boldsymbol{\omega}^{\star}\right)-\bar{h}\left(\boldsymbol{x}\right)\right|$$

$$\leq\lambda\|M_{K_h,\bar{K}}\|_2+8\sqrt{\left(1+\lambda\|M_{K_h,\bar{K}}\|_2\right)\|M_{K_h,\bar{K}}\|_2}+8\frac{\sqrt{Dn}}{D\lambda+n}+6\sqrt{\frac{\log\frac{4}{\delta}}{2n}}, \tag{42}$$

*where the empirical difference $\frac{1}{n}\sum_{i=1}^{n}\left|h\left(\boldsymbol{x}_i;\boldsymbol{\omega}^{\star}\right)-\bar{h}\left(\boldsymbol{x}_i\right)\right|$ is upper bounded by $\lambda\|M_{K_h,\bar{K}}\|_2$.*

*Proof.* Our goal is to bound the right-hand side of Eq. (30) using the $L_2$-geometric difference $\|M_{K_h,\bar{K}}\|_2$ between the two kernel matrices. First, we calculate its first term which is the upper bound of the empirical difference $\frac{1}{n}\sum_{i=1}^{n}\left|h\left(\boldsymbol{x}_i;\boldsymbol{\omega}^{\star}\right)-\bar{h}\left(\boldsymbol{x}_i\right)\right|$.

It is straightforward to calculate that

$$\frac{1}{n}\left\|K_h(K_h+\lambda I)^{-1}Y-\frac{1}{D\lambda+n}JY\right\|_1=\frac{1}{n}\left\|K_h(K_h+\lambda I)^{-1}Y-\bar{K}\left(\bar{K}+\lambda I\right)^{-1}Y\right\|_1$$

$$=\frac{\lambda}{n}\left\|(K_h+\lambda I)^{-1}Y-\left(\bar{K}+\lambda I\right)^{-1}Y\right\|_1 \tag{43}$$

$$\leq\frac{\lambda}{\sqrt{n}}\left\|(K_h+\lambda I)^{-1}Y-\left(\bar{K}+\lambda I\right)^{-1}Y\right\|_2 \tag{44}$$

$$\leq\frac{\lambda}{\sqrt{n}}\|M_{K_h,\bar{K}}\|_2\|Y\|_2 \tag{45}$$

$$\leq\lambda\|M_{K_h,\bar{K}}\|_2, \tag{46}$$

where Eq. (43) uses $K(K+\lambda I)^{-1}=I-\lambda(K+\lambda I)^{-1}$, Eq. (44) comes from the fact that for an $n$-dimensional vector $\boldsymbol{x}$, $\|\boldsymbol{x}\|_1\leq\sqrt{n}\|\boldsymbol{x}\|_2$, Eq. (45) employs $\|AY\|_2\leq\|A\|_2\|Y\|_2$, and Eq. (46) utilizes the fact that $\|Y\|_2\leq\sqrt{n}$ owing to $\|O\|_2\leq1$.

To upper bound the second term in the right-hand side of Eq. (30), we decompose it into two terms and employ the triangle inequality to yield

$$\frac{8}{\sqrt{n}}\left[\sqrt{Y^{\mathrm{T}}(K_h+\lambda I)^{-1}K_h(K_h+\lambda I)^{-1}Y}\right]$$

$$\leq\frac{8}{\sqrt{n}}\left[\sqrt{Y^{\mathrm{T}}(K_h+\lambda I)^{-1}K_h(K_h+\lambda I)^{-1}Y-Y^{\mathrm{T}}\left(\bar{K}+\lambda I\right)^{-1}\bar{K}\left(\bar{K}+\lambda I\right)^{-1}Y}\right]$$

$$+\frac{8}{\sqrt{n}}\left[\sqrt{Y^{\mathrm{T}}\left(\bar{K}+\lambda I\right)^{-1}\bar{K}\left(\bar{K}+\lambda I\right)^{-1}Y}\right]. \tag{47}$$

Since it can be verified that

$$\left(\bar{K}+\lambda I\right)^{-1}\bar{K}\left(\bar{K}+\lambda I\right)^{-1}=\frac{D}{\left(D\lambda+n\right)^2}J,$$

we have

$$\sqrt{Y^{\mathrm{T}}\left(\bar{K}+\lambda I\right)^{-1}\bar{K}\left(\bar{K}+\lambda I\right)^{-1}Y}=\sqrt{Y^{\mathrm{T}}\frac{D}{\left(D\lambda+n\right)^2}JY}=\frac{\sqrt{D}}{D\lambda+n}\left|\sum_{i=1}^{n}y_i\right|\leq\frac{\sqrt{D}n}{D\lambda+n}, \tag{48}$$

where Eq. (48) utilizes $\left|\sum_{i=1}^{n}y_i\right|\leq n$.

Moreover, by employing $K(K + \lambda I)^{-1} = I - \lambda(K + \lambda I)^{-1}$, it can be calculated that

$$
(K_h + \lambda I)^{-1} K_h (K_h + \lambda I)^{-1} - (\bar{K} + \lambda I)^{-1} \bar{K} (\bar{K} + \lambda I)^{-1}
$$

$$
= (K_h + \lambda I)^{-1} \left[ K_h (K_h + \lambda I)^{-1} - \bar{K}(\bar{K} + \lambda I)^{-1} \right]
$$

$$
+ \left[ (K_h + \lambda I)^{-1} - (\bar{K} + \lambda I)^{-1} \right] \bar{K}(\bar{K} + \lambda I)^{-1}
$$

$$
= - \lambda (K_h + \lambda I)^{-1} \left[ (K_h + \lambda I)^{-1} - (\bar{K} + \lambda I)^{-1} \right]
$$

$$
+ \left[ (K_h + \lambda I)^{-1} - (\bar{K} + \lambda I)^{-1} \right] \left[ I - \lambda (\bar{K} + \lambda I)^{-1} \right]
$$

$$
= \left[ (K_h + \lambda I)^{-1} - (\bar{K} + \lambda I)^{-1} \right] - \lambda \left[ (K_h + \lambda I)^{-2} - (\bar{K} + \lambda I)^{-2} \right]
$$

$$
= M_{K_h, \bar{K}} - \lambda \left[ M_{K_h, \bar{K}}^2 + M_{K_h, \bar{K}} (\bar{K} + \lambda I)^{-1} + (\bar{K} + \lambda I)^{-1} M_{K_h, \bar{K}} \right],
$$

where $M_{K_h, \bar{K}} = (K_h + \lambda I)^{-1} - (\bar{K} + \lambda I)^{-1}$. Thus, the corresponding quadratic form reads

$$
Y^{\mathrm{T}} (K_h + \lambda I)^{-1} K_h (K_h + \lambda I)^{-1} Y - Y^{\mathrm{T}} (\bar{K} + \lambda I)^{-1} \bar{K} (\bar{K} + \lambda I)^{-1} Y
$$

$$
= Y^{\mathrm{T}} M_{K_h, \bar{K}} Y - \lambda Y^{\mathrm{T}} \left[ M_{K_h, \bar{K}}^2 + M_{K_h, \bar{K}} (\bar{K} + \lambda I)^{-1} + (\bar{K} + \lambda I)^{-1} M_{K_h, \bar{K}} \right] Y
$$

$$
= Y^{\mathrm{T}} \left[ I - \lambda M_{K_h, \bar{K}} - 2\lambda (\bar{K} + \lambda I)^{-1} \right] M_{K_h, \bar{K}} Y
$$

$$
\leq \left\| \left[ I - \lambda M_{K_h, \bar{K}} - 2\lambda (\bar{K} + \lambda I)^{-1} \right] M_{K_h, \bar{K}} \right\|_2 \|Y\|_2^2 \tag{49}
$$

$$
\leq n \left[ \left\| I - 2\lambda (\bar{K} + \lambda I)^{-1} \right\|_2 + \lambda \|M_{K_h, \bar{K}}\|_2 \right] \|M_{K_h, \bar{K}}\|_2 \tag{50}
$$

$$
\leq n \left( 1 + \lambda \|M_{K_h, \bar{K}}\|_2 \right) \|M_{K_h, \bar{K}}\|_2, \tag{51}
$$

where Eq. (49) uses $Y^{\mathrm{T}} A Y \leq \|A\|_2 \|Y\|_2^2$, Eq. (50) employs the triangle inequality and the sub-multiplicative property of matrix norm as well as the inequality that $\|Y\|_2 \leq \sqrt{n}$, and Eq. (51) utilizes $\left\| I - 2\lambda (\bar{K} + \lambda I)^{-1} \right\|_2 = 1$ which can be verified by checking the maximum singular value of $I - 2\lambda (\bar{K} + \lambda I)^{-1}$.

By plugging Eqs. (51) and (48) into Eq. (47), it yields

$$
\frac{8}{\sqrt{n}} \left[ \sqrt{Y^{\mathrm{T}} (K_h + \lambda I)^{-1} K_h (K_h + \lambda I)^{-1} Y} \right] \leq 8\sqrt{\left( 1 + \lambda \|M_{K_h, \bar{K}}\|_2 \right) \|M_{K_h, \bar{K}}\|_2} + 8 \frac{\sqrt{Dn}}{D\lambda + n}.
\tag{52}
$$

Thus, combining Eqs. (46), (52) with (30), we have the conclusion of Lemma B.2. $\qquad\square$

The proof of Lemma 3.2 can be completed by letting the hypothesis function $h(\boldsymbol{x}; \boldsymbol{\omega}^{\star})$ be $\tilde{h}(\boldsymbol{x})$ and the associated kernel matrix $K_h$ be $\widetilde{K}$ in Lemma B.2.

## C  PROOF OF THEOREM 3.1

*Proof.* It is straightforward to verify that by plugging Eq. (22) into Eq. (21) and further simplifying the resultant equation, Theorem 3.1 can be proved. Now we provide the proof of Eq. (22), namely,

$$
\|M_{\widetilde{K}, \bar{K}}\|_2 \leq \frac{\frac{n}{\lambda^2} (1 - p) \left( 1 + \frac{1}{D} \right)}{1 - \frac{n}{\lambda} (1 - p) \left( 1 + \frac{1}{D} \right)}.
$$

The proof leverages the following lemma.

**Lemma C.1.** *(Lemma 6, Wang et al. (2021)) Let $\| \cdot \|$ be a given matrix norm and suppose $A, B \in \mathbb{R}^{n \times n}$ are nonsingular and satisfy $\|A^{-1}(A - B)\| \leq 1$, then*

$$
\|A^{-1} - B^{-1}\| \leq \frac{\|A^{-1}\|^2 \|A - B\|}{1 - \|A^{-1}(A - B)\|}.
\tag{53}
$$

From Lemma C.1, we have

$$
\begin{aligned}
\|M_{\widetilde{K},\bar{K}}\|_2 &= \left\| \left(\bar{K}+\lambda I\right)^{-1} - \left(\widetilde{K}+\lambda I\right)^{-1} \right\|_2 \\
&\leq \frac{\left\|\left(\bar{K}+\lambda I\right)^{-1}\right\|_2^2 \|\bar{K}-\widetilde{K}\|_2}{1-\left\|\left(\bar{K}+\lambda I\right)^{-1}\left(\bar{K}-\widetilde{K}\right)\right\|_2} \\
&\leq \frac{\left\|\left(\bar{K}+\lambda I\right)^{-1}\right\|_2^2 \|\bar{K}-\widetilde{K}\|_2}{1-\left\|\left(\bar{K}+\lambda I\right)^{-1}\right\|_2 \|\bar{K}-\widetilde{K}\|_2} & (54) \\
&\leq \frac{\frac{n}{\lambda^2}(1-p)\left(1+\frac{1}{D}\right)}{1-\frac{n}{\lambda}(1-p)\left(1+\frac{1}{D}\right)}, & (55)
\end{aligned}
$$

where Eq. (54) uses the sub-multiplicative property of matrix norm, and Eq. (55) employs the facts that $\|\left(\bar{K}+\lambda I\right)^{-1}\|_2 = \frac{1}{\lambda}$ and

$$
\begin{aligned}
\|\bar{K}-\widetilde{K}\|_2 &= (1-p)\,\|K-\bar{K}\|_2 \\
&\leq (1-p)\left(\|K\|_2 + \|\bar{K}\|_2\right) \\
&\leq n(1-p)\left(1+\frac{1}{D}\right). & (56)
\end{aligned}
$$

Here, we have utilized $\|K\|_2 \leq \mathrm{Tr}(K) \leq n$ and $\|\bar{K}\|_2 = \frac{n}{D}$. $\qquad\square$

## D  PROOF OF COROLLARY 3.4

*Proof.* According to the assumption of balanced labels and the Hoeffding's inequality (Lemma A.1), for any $\epsilon>0$, we have

$$
\mathbb{P}\left(\left|\frac{1}{n}\sum_{i=1}^n y_i\right| \geq \epsilon\right) \leq 2e^{-n\epsilon^2/2}. \tag{57}
$$

Thus, for any $\delta_1>0$, with probability of at least $1-\delta_1$ over the draw of $S$, it holds that

$$
\left|\frac{1}{n}\sum_{i=1}^n y_i\right| \leq \sqrt{\frac{2\log\frac{2}{\delta_1}}{n}}, \tag{58}
$$

so that

$$
\sqrt{Y^{\mathrm{T}}\left(\bar{K}+\lambda I\right)^{-1}\bar{K}\left(\bar{K}+\lambda I\right)^{-1}Y} = \frac{\sqrt{D}}{D\lambda+n}\left|\sum_{i=1}^n y_i\right| \leq \frac{\sqrt{Dn}}{D\lambda+n}\sqrt{2\log\frac{2}{\delta_1}}. \tag{59}
$$

This can yield a tighter bound of the second term in the right-hand side of Eq. (47) than the bound in Eq. (48).

In fact, by replacing Eq. (48) with Eq. (59) in Eq. (20) and employing the sub-additivity of probability, we derive that for any $\delta_1,\delta_2>0$, with probability of at least $1-\delta_1-\delta_2$ over the draw of $S$,

$$
\mathbb{E}_{(\boldsymbol{x},y)\sim\mathcal{D}}\left|\tilde{h}\left(\boldsymbol{x}\right) - \bar{h}\left(\boldsymbol{x}\right)\right| \leq f\left(\frac{n}{\lambda}(1-p)\left(1+\frac{1}{D}\right)\right) + \frac{8\sqrt{D}}{D\lambda+n}\sqrt{2\log\frac{2}{\delta_1}} + 6\sqrt{\frac{\log\frac{4}{\delta_2}}{2n}}. \tag{60}
$$

Finally, letting $\delta_1 = \delta_2 = \frac{\delta}{2}$, we prove Corollary 3.4. $\qquad\square$

## E  PROOF OF THEOREM 3.5

Before giving the proof of Theorem 3.5, we first provide a necessary lemma to guarantee the existence of $\hat{h}\left(\boldsymbol{x}\right)$, the optimal hypothesis inferred by the estimated noisy kernel.

**Lemma E.1.** *With probability of at least $1 - ne^{-\lambda^2 m/4n}$, the estimated noisy kernel matrix $\widehat{K}$ satisfies*

$$\widehat{K} + \frac{\lambda}{2}I \succeq \widetilde{K} \succeq 0, \tag{61}$$

*where $\widetilde{K}$ is the noisy kernel matrix.*

*Proof.* According to our settings, for any $i, j \in [n]$, we have

$$\widehat{\mathcal{K}}\left(\boldsymbol{x}_i, \boldsymbol{x}_j\right) = \frac{1}{m}\sum_{k=1}^{m} V_k\left(\boldsymbol{x}_i, \boldsymbol{x}_j\right), \tag{62}$$

where each $V_k\left(\boldsymbol{x}_i, \boldsymbol{x}_j\right) \triangleq V_{k;ij}$ is a Bernoulli random variable with expectation $\widetilde{\mathcal{K}}\left(\boldsymbol{x}_i, \boldsymbol{x}_j\right) = \widetilde{K}_{ij}$. For any $i, j \in [n]$, $k \in [m]$, let

$$Y^{(k;ij)} = \frac{1}{m}\left(V_{k;ij} - \widetilde{K}_{ij}\right)E^{(ij)}, \tag{63}$$

where $E^{(ij)} = |i\rangle\langle j| + |j\rangle\langle i|$, and particularly, $E^{(ii)} = 2|i\rangle\langle i|$. It is clear that the expectation of the random Hermitian $n \times n$ matrix $Y^{(k;ij)}$ is zero and

$$
\begin{aligned}
\left(Y^{(k;ij)}\right)^2 &= \frac{1}{m^2}\left(V_{k;ij} - \widetilde{K}_{ij}\right)^2\left(E^{(ij)}\right)^2 \\
&= \frac{1}{2m^2}\left(V_{k;ij} - \widetilde{K}_{ij}\right)^2\left(E^{(ii)} + E^{(jj)}\right) \\
&\preceq \frac{1}{2m^2}\left(E^{(ii)} + E^{(jj)}\right),
\end{aligned}
\tag{64}
$$

where Eq. (64) is derived from the inequality of $\left(V_{k;ij} - \widetilde{K}_{ij}\right)^2 \leq 1$.

According to Lemma A.2, for all $t \geq 0$,

$$\mathbb{P}\left[\sum_{i,j=1}^{n}\sum_{k=1}^{m} Y^{(k;ij)} + tI \succeq 0\right] \geq 1 - ne^{-t^2/2\sigma^2}, \tag{65}$$

with

$$
\begin{aligned}
\sigma^2 &= \frac{1}{2}\left\|\sum_{i,j=1}^{n}\sum_{k=1}^{m}\left[\frac{1}{2m^2}\left(E^{(ii)} + E^{(jj)}\right) + \mathbb{E}\left(Y^{(k;ij)}\right)^2\right]\right\|_2 \\
&= \frac{1}{2}\left\|\sum_{i,j=1}^{n}\left[\frac{1}{2m}\left(E^{(ii)} + E^{(jj)}\right) + \frac{1}{2m^2}\sum_{k=1}^{m}\mathbb{E}\left(V_{k;ij} - \widetilde{K}_{ij}\right)^2\left(E^{(ii)} + E^{(jj)}\right)\right]\right\|_2 \\
&\leq \frac{1}{2m}\left\|\sum_{i,j=1}^{n}\left(E^{(ii)} + E^{(jj)}\right)\right\|_2 \\
&= \frac{1}{2m}\|4nI\|_2 = \frac{2n}{m}.
\end{aligned}
\tag{66}
$$

Here, Eq. (66) is derived from the inequality of $\left(V_{k;ij} - \widetilde{K}_{ij}\right)^2 \leq 1$.

Thus, by letting $t = \lambda$ and noting that

$$\sum_{i,j=1}^{n}\sum_{k=1}^{m} Y^{(k;ij)} = \sum_{i,j=1}^{n}\left(\widehat{K}_{ij} - \widetilde{K}_{ij}\right)E^{(ij)} = 2\left(\widehat{K} - \widetilde{K}\right), \tag{67}$$

we have

$$\mathbb{P}\left[\widehat{K} + \frac{\lambda}{2}I \succeq \widetilde{K}\right] \geq 1 - ne^{-\lambda^2 m/4n}, \tag{68}$$

which completes the proof. $\square$

From Lemma E.1, the positive definiteness of $\widehat{K} + \lambda I$ guarantees that $\hat{h}(\boldsymbol{x})$ exists and can be described in the form of Eq. (19). Now we present the proof of Theorem 3.5.

*Proof.* According to Lemma C.1, we have

$$
\begin{aligned}
\|M_{\widehat{K}, \bar{K}}\|_2 &= \left\| \left(\bar{K} + \lambda I\right)^{-1} - \left(\widehat{K} + \lambda I\right)^{-1} \right\|_2 \\
&\leq \frac{\left\|\left(\bar{K} + \lambda I\right)^{-1}\right\|_2^2 \|\bar{K} - \widehat{K}\|_2}{1 - \left\|\left(\bar{K} + \lambda I\right)^{-1}\left(\bar{K} - \widehat{K}\right)\right\|_2} \\
&\leq \frac{\frac{1}{\lambda^2}\|\bar{K} - \widehat{K}\|_2}{1 - \frac{1}{\lambda}\|\bar{K} - \widehat{K}\|_2},
\end{aligned}
\tag{69}
$$

where Eq. (69) employs the sub-multiplicative property of matrix norm and $\left\|\left(\bar{K} + \lambda I\right)^{-1}\right\|_2 = \frac{1}{\lambda}$. Moreover,

$$
\begin{aligned}
\|\bar{K} - \widehat{K}\|_2 &\leq \|\bar{K} - \widetilde{K}\|_2 + \|\widetilde{K} - \widehat{K}\|_2 \\
&\leq n(1-p)\left(1 + \frac{1}{D}\right) + \|\widetilde{K} - \widehat{K}\|_2,
\end{aligned}
\tag{70}
$$

where Eq. (70) is derived from Eq. (56).

According to the definition of $\widehat{\mathcal{K}}(\boldsymbol{x}, \boldsymbol{x}')$ in Eq. (18) and the Hoeffding's inequality (Lemma A.1), for any $\epsilon \geq 0$ and arbitrary $\boldsymbol{x}, \boldsymbol{x}' \in \mathcal{X}$, we have

$$
\mathbb{P}\left(\left|\widehat{\mathcal{K}}(\boldsymbol{x}, \boldsymbol{x}') - \widetilde{\mathcal{K}}(\boldsymbol{x}, \boldsymbol{x}')\right| \geq \epsilon\right) \leq 2e^{-2\epsilon^2 m}.
\tag{71}
$$

Note that the estimated noisy kernel matrix $\widehat{K}$ is a random matrix with its expectation being the noisy kernel matrix $\widetilde{K}$. Then for any $\epsilon \geq 0$,

$$
\begin{aligned}
\mathbb{P}\left(\left\|\widetilde{K} - \widehat{K}\right\|_2 \geq \epsilon\right) &\leq \mathbb{P}\left(\left\|\widetilde{K} - \widehat{K}\right\| \geq \epsilon\right) \\
&= \mathbb{P}\left(\sum_{i=1}^{n}\sum_{j=1}^{n}\left|\widetilde{K}_{ij} - \widehat{K}_{ij}\right|^2 \geq \epsilon^2\right) \\
&\leq \mathbb{P}\left(\bigcup_{i=1}^{n}\bigcup_{j=1}^{n}\left\{\left|\widetilde{K}_{ij} - \widehat{K}_{ij}\right|^2 \geq \frac{\epsilon^2}{n^2}\right\}\right) \\
&\leq \sum_{i=1}^{n}\sum_{j=1}^{n}\mathbb{P}\left(\left|\widetilde{K}_{ij} - \widehat{K}_{ij}\right|^2 \geq \frac{\epsilon^2}{n^2}\right) \\
&\leq 2n^2 e^{-2m\epsilon^2/n^2},
\end{aligned}
\tag{72, 73}
$$

where Eq. (72) employs the relationship between the spectral norm and the Frobenius norm, namely, $\|A\|_2 \leq \|A\|$, and Eq. (73) is obtained from Eq. (71).

Combining Eqs. (69), (70) and (73), it yields that for any $\delta_1 > 0$, with probability of at least $1 - \delta_1$,

$$
\|M_{\widehat{K}, \bar{K}}\|_2 \leq \frac{\frac{1}{\lambda^2}\left[n(1-p)\left(1 + \frac{1}{D}\right) + \sqrt{\frac{n^2}{2m}\log\frac{2n^2}{\delta_1}}\right]}{1 - \frac{1}{\lambda}\left[n(1-p)\left(1 + \frac{1}{D}\right) + \sqrt{\frac{n^2}{2m}\log\frac{2n^2}{\delta_1}}\right]}.
\tag{74}
$$

Thus, from Lemma E.1, Lemma 3.2, and Eq. (74), by employing the sub-additivity of probability, we have the conclusion that for any $\delta_1, \delta_2 > 0$, with probability of at least $1 - \delta_1 - \delta_2 - ne^{-\lambda^2 m/4n}$,

$$
\mathbb{E}_{(\boldsymbol{x}, y) \sim \mathcal{D}}\left|\hat{h}(\boldsymbol{x}) - \bar{h}(\boldsymbol{x})\right| \leq f\left(\frac{n}{\lambda}(1-p)\left(1 + \frac{1}{D}\right) + \frac{n}{\lambda}\sqrt{\frac{\log\frac{2n^2}{\delta_1}}{2m}}\right) + \frac{8\sqrt{Dn}}{D\lambda + n} + 6\sqrt{\frac{\log\frac{4}{\delta_2}}{2n}},
$$

where $f(z) = \frac{z + 8\sqrt{\frac{z}{\lambda}}}{1 - z}$.

Finally, by letting $\delta_1 = \delta_2 = \frac{\delta}{2}$, we reach the conclusion of Theorem 3.5. $\qquad\square$

