# OpenReview forum: "Power Characterization of Noisy Quantum Kernels"
_ICLR.cc/2024/Conference — Submitted to ICLR 2024_

### Official Review · Reviewer_XjgB · 2023-10-18

**Soundness:** 3 good
**Presentation:** 2 fair
**Contribution:** 2 fair
**Rating:** 5
**Confidence:** 4

**Summary:**

The noise tolerance of the quantum kernel method. The noise model considered in this paper is to apply a global depolarizing channel to each layer of the quantum encoding circuit. This paper shows a theoretical characterization of the prediction performance of the quantum kernel in terms of the number of training data, the number of qubits $N$, the strength of the noise $\tilde{p}$, and the number of layers. It mainly studies three regimes of the training data size. For logarithmically small data, the kernel always fails. For $poly(N)$ size data, it fails when the number of layers is $\Omega(\log (N)/\log (1-p))$. And for $\exp(N)$ size data, the kernel fails when the number of layers is $\Omega(N/\log(1-p))$. Technically, it bounds the L1 distance between the hypothesis obtained from a noisy kernel and a constant function via the Rademacher complexity.

**Strengths:**

The problem investigated in this paper is a crucial and foundational problem in the field of quantum machine learning. In comparison with previous findings, the noise model considered in this paper is less restrictive. Furthermore, the result holds for quantum circuits of any depth and width and does not rely on strong constraints on the circuit architecture. It reveals some limitations in QML, especially in the NISQ era. Most of the theorems/lemmas appear to be mathematically sound to me.

**Weaknesses:**

This paper does not clearly state its differences from prior works, particularly in terms of technical detail. Moreover, most of the results are derived from classical kernel theory. Additionally, the result itself is not surprising. Depolarizing channels can be equivalent to a single depolarizing channel with noise strength that grows exponentially close to 1 as the number of layers increases. Therefore, the output state of the encoding circuit will converge to the maximally mixed state as the number of layers increases. As such, the characterization derived in this paper is quite natural and intuitive. This paper could be strengthened by considering more complicated and practical noise models, as well as the effect of error mitigation. It would be beneficial to conduct further experiments (e.g., testing on real quantum devices).

**Questions:**

Do you have experiments showing the test errors and generalization errors for different numbers of training data (fixing other parameters)?

Line -5 before Sec. 1.3: “...required for probably successful training”, what does “probably successful” mean?

Page 2, line -7: what is $q$ in “$q^N$”? Is it a circuit parameter or any constant?

Page 3: “indicate that good generalization alone does not necessarily guarantee good prediction for new data”. Does “good generalization” mean the generalization of the noiseless kernel?

Eq. (3): it is unclear whether the square is inside or outside of the expectation.

Sec. 2.2: the introduction to the quantum kernel method is not self-contained. A complete workflow should be given for the ease of readers who are unfamiliar with this field. And it should mention which part is quantum and which part is classical.

Page 7, line 5: “The depolarization noise model in Theorem 3.1 is weaker than the noise model…” Weaker in what sense? Could you state it more explicitly?

---

> ### Author Response · Authors · 2023-11-17
> **Response to Reviewer XjgB (weaknesses)**
>
> Thank you very much for your helpful comments and suggestions.
>
> Firstly, following your constructive advice, we have thoroughly revised Sec. 1.2 Related Work to present our contributions and the differences from prior works more clearly.
>
> Secondly, although the result itself may not be very surprising, we do explicitly depict the speed of the prediction concentration of noisy quantum kernels in terms of the strength of depolarization noise, the size of training samples, the number of qubits, the number of layers affected by quantum noise, and the number of measurement shots. This helps us quantitatively understand the limits of noisy quantum kernel methods. As an illustration, for a training sample set, whose scale is exponential in the number of qubits $N$, like $q^N$ for some $q>1$, noisy quantum kernel methods fail once the number of noisy layers exceeds $N\log\_{(1−\tilde{p} )^{−2}} q$.
>
> Thirdly, when presenting negative results concerning noisy quantum kernels, we feel it is better to assume a relatively weaker noise model. Once the kernel methods fail under weaker noise, they fail under stronger ones in general. In the reply to your last question, we detailedly explain that our global noise model is actually weaker than the noise models considered in previous works. In addition, the noise rate $\tilde{p}>0$ in our work can be arbitrarily small. Therefore, it can depict the case where the noise influence is very weak, namely, after the noise the quantum state is left untouched with a very high probability $1-\tilde{p}$. Thus, in this sense, our global depolarizing model is an appropriate starting point to study the limitations of noisy quantum kernel methods. We have made this clearer in the revised manuscript after Eq. (12).
>
> We agree with you that much work needs to be done to further investigate the limitations of quantum kernel methods. For example, consider more realistic quantum noise including the effect of mitigation, and conduct experiments on real quantum devices. We leave them for future research. Thank you for pointing out these potential research directions.

---

> ### Author Response · Authors · 2023-11-17
> **Response to Reviewer XjgB (Questions)**
>
> - Do you have experiments showing the test errors and generalization errors for different numbers of training data (fixing other parameters)?
>
> Yes, we also consider the case where the number of training data is 100, which is polynomial in the number of qubits $N$. In this case the numerical results also match well with our theoretical results.
>
> - Line -5 before Sec. 1.3: “...required for probably successful training”, what does “probably successful” mean?
>
> We mean that for those cases, to have a good training, a necessary condition is that exponentially many resources are required. However, this is not a sufficient condition, namely, exponential number of resources do not necessarily guarantee a good training under those scenarios. In the revised paper, we have removed  “probably successful” to avoid possible confusion.
>
> - Page 2, line -7: what is $q$ in $q^N$? Is it a circuit parameter or any constant?
>
> It is one of the typical sizes of the training sample we consider. To be specific, $q^N$ with $q>1$ denotes that the size of the training sample is exponentially large in the number of qubits $N$. We have made this clearer in the revised version.
>
> - Page 3: “indicate that good generalization alone does not necessarily guarantee good prediction for new data”. Does “good generalization” mean the generalization of the noiseless kernel?
>
> Here, “good generalization” refers to the generalization for general quantum machine learning methods. Recall that generalization error is the difference between the prediction error and the training error. When both the training and prediction are bad, its generalization error may still be small. Thus, for any quantum machine learning methods, only having good generalization cannot necessarily guarantee good prediction. To have good prediction, both the training and generalization errors should be small. We have revised the paper to make it clearer.
>
> - Eq. (3): it is unclear whether the square is inside or outside of the expectation.
>
> We have revised the expression to make it clearer. Thank you for pointing out the potential confusion.
>
> - Sec. 2.2: the introduction to the quantum kernel method is not self-contained. A complete workflow should be given for the ease of readers who are unfamiliar with this field. And it should mention which part is quantum and which part is classical.
>
> We have revised Sec. 2.2 to make it more self-contained. Specifically, we have clearly pointed out which part is quantum and which part is classical. Thank you for your suggestion.
>
> - Page 7, line 5: “The depolarization noise model in Theorem 3.1 is weaker than the noise model…” Weaker in what sense? Could you state it more explicitly?
>
> Thank you for your constructive comment. We have added more explanations on “weaker” in what sense in the revised manuscript. In one word, our noise model is weaker in the sense that with the same depolarizing rate $\tilde{p}$, under our noise model the quantum state is left untouched with an exponentially larger probability $(1-\tilde{p})$ as compared to that $(1-\tilde{p})^N$ under the noise model considered in previous works. Here, $N$ denotes the number of qubits.
>
> Specifically, in our work, the noise model, $\mathcal{N}\_{\tilde{p}}=(1-\tilde{p})\rho+\tilde{p}\frac{1}{D}I$ with rate $\tilde{p}$, is referred to as “global” as it applies onto all qubits after each layer of ideal unitary gates. While in Stilck Franca & Garcia-Patron (2021) and De Palma et al. (2023), the considered noise model is the local depolarizing noise. However, it applies to each qubit. Thus, their noise channel applied after each layer of unitary gates is $\mathcal{D}\_{\tilde{p}}^{\otimes N}$, with $\mathcal{D}\_{\tilde{p}}=(1-\tilde{p})\rho+\frac{1}{2}I$ and $N$ being the number of qubits. We have rewritten it using our notation for convenience of comparison. With the same noise rate $\tilde{p}$, we can compare our global depolarizing noise $\mathcal{N}\_{\tilde{p}}$ with its counterpart $\mathcal{D}\_{\tilde{p}}^{\otimes N}$. It can be seen that under our global noise, the quantum state is left untouched with probability $1-\tilde{p}$, while under the so-called local noise, the probability of the state remaining unchanged is only $(1-\tilde{p})^N$, which is exponentially small in $N$. Thus, in this sense our global depolarizing model is weaker than the so-called local depolarizing model. As for the Pauli noise model considered in Thanasilp et al. (2022), following a similar analysis, we can see that the Pauli noise is also stronger than ours. We have made this clearer in the revised manuscript after Eq. (12).

---

> ### Author Response · Authors · 2023-11-22
>
> Dear Reviewer XjgB,
>
> Many thanks for your dedicated time to review our work. We have taken your insightful comments seriously and have made necessary modifications based on your constructive suggestions. With the imminent closure of the discussion period, time is of the essence.
>
> Given that our work received your’s less favorable scores, we worked diligently to clarify the difference from prior works and answer your specific questions during the rebuttal period, demonstrating our unwavering commitment to addressing every facet of your feedback.
>
> We kindly request your esteemed attention to our rebuttal response. Your responsible and positive endorsement holds paramount significance during this critical phase.
>
> Wishing you all the best and awaiting your valued evaluation.
>
> Best regards,
>
> Authors

---

> > ### Comment · Reviewer_XjgB · 2023-11-22
> >
> > We thank the authors for your detailed response. However, I’m still concerned about the technical contribution of this paper. And I also agree with other reviewers about the local and global depolarizing noises issue. Therefore, I’ll keep my score unchanged.

---

> ### Author Response · Authors · 2023-11-22
> **New results on applicability to local depolarizing noises**
>
> Thank you so much for your feedback. We have worked diligently for a continuous span of ten days during the rebuttal period to explore this issue on local depolarizing noise. Finally, we are excited to share with you that we find our main result can be pushed to local depolarizing noise so that the prediction difference $\vert \tilde{h}-\bar{h} \vert$ converges to zero with the circuit depth increasing. Specifically, we outline the additional proof for local depolarizing noises as follows.
>
> Firstly, after 1-layer of $N$-fold local depolarizing channel, the relative entropy satisfies  $S(\mathcal{D}\_{\tilde{p}}^{\otimes N}(\rho\_{0})\Vert \frac{1}{D}I)\leq (1-\tilde{p})^2 S(\rho\_{0}\Vert \frac{1}{D}I)$ [1].
>
> Secondly, similar to Eq. (14) in our paper, denote by $\rho\_{\mathcal{D}}(x; x\_{i})$ the final state of the $2L$-layer local depolarizing noisy quantum circuit. Then, the corresponding kernel value is $\mathcal{K}(x,x\_{i})={\rm Tr} \left\[ P\_{0} \ \rho\_{\mathcal{D}}(x; x\_{i})\right\]$.
> For local-depolarizing noise, we have $\vert \mathcal{K}(x,x\_{i}) - \overline{\mathcal{K}}(x,x\_{i}) \vert \leq \Vert \rho\_{\mathcal{D}}(x; x\_{i})-\frac{1}{D}I \Vert\_{1} \leq  \sqrt{2 \ln 2\ S(\rho\_{\mathcal{D}}(x; x\_{i})\Vert\frac{1}{D}I)} \leq (1-\tilde{p})^{2L}\sqrt{2 \ln 2\ S(\rho\_{0}\Vert\frac{1}{D}I)}$, where the first two inequality come from the matrix H$\ddot{\rm o}$lder Inequality and quantum Pinsker Inequality [2], respectively. Thus, $\vert h(x)-\bar{h}(x) \vert = \mathcal{O}(n\vert \mathcal{K}(x,x\_{i}) - \overline{\mathcal{K}}(x,x\_{i}) \vert)=\mathcal{O}(n(1-\tilde{p})^{2L})$, where $n$ is the sample size.
>
> Thirdly, for our global depolarizing noise, since $S(\mathcal{N}\_{\tilde{p}} (\rho\_{0})\Vert \frac{1}{D}I)\leq (1-\tilde{p}) S(\rho\_{0}\Vert \frac{1}{D}I)$, we have $\vert h(x)-\bar{h}(x) \vert = \mathcal{O}(n(1-\tilde{p})^{L})$ by following a similar analysis.
>
> Combing the above statements, we can find that after local depolarizing noise, the kernel tends to get worse even for shallow circuits. The fundamental reason is that after an 1-layer of global depolarizing channel, the relative entropy decays as $S(\mathcal{N}\_{\tilde{p}} (\rho\_{0})\Vert \frac{1}{D}I)\leq (1-\tilde{p}) S(\rho\_{0}\Vert \frac{1}{D}I)$, while for the local depolarizing noise $S(\mathcal{D}\_{\tilde{p}}^{\otimes N}(\rho\_{0})\Vert \frac{1}{D}I)\leq (1-\tilde{p})^2 S(\rho\_{0}\Vert \frac{1}{D}I)$, which decays in a faster way.
>
> In the current quantum devices, the noise is vital of consideration before realizing quantum error correction and fault tolerant quantum computing. To this end, we and the authors in many Related Works are focusing on the effect of quantum noise and provide a crucial warning to utilize noisy quantum kernels under different noise models.
>
> In a word, for the ultimate goal of machine learning, i.e., prediction, the local depolarizing noise model is usually “stronger” than our global one. Our work is especially useful for developing quantum kernel methods in noisy intermediate-scale quantum (NISQ) era where we have no sufficient capability for quantum error correction. We agree with you that further work is worthing being presented to discuss the different performance and effects of quantum error correction and quantum error mitigation for various noise models.
>
> [1] Stilck França, D., García-Patrón, R. Limitations of optimization algorithms on noisy quantum devices. Nat. Phys. 17, 1221–1227 (2021). https://doi.org/10.1038/s41567-021-01356-3
>
> [2] Watrous, J. The Theory of Quantum Information. Cambridge: Cambridge University Press. https://doi.org/10.1017/9781316848142, Theorem 5.38 for quantum Pinsker Inequality.
>
> We do expect that our further clarification addresses your concerns. Nevertheless, the final decision for endorsement or rejection ultimately lies with you, and we deeply respect that.
>
> Many thanks for your advice, suggestions and feedback, and wish you all the best.
>
> Best regards,
>
> Authors

---

### Official Review · Reviewer_M7Gm · 2023-10-19

**Soundness:** 4 excellent
**Presentation:** 3 good
**Contribution:** 2 fair
**Rating:** 6
**Confidence:** 3

**Summary:**

The authors examine the noise-resilience of quantum kernel methods. To achieve this, they first consider the performance of a quantum kernel method $\overline{h}$ completely dominated by depolarizing noise. They then bound the expected difference in prediction between $\overline{h}$ and a kernel method at a fixed rate of depolarizing noise, and study when this expected difference vanishes. They consider when the training sample size grows logarithmically, polynomially, and exponentially with the number of qubits used in the quantum kernel, and show that at a fixed depolarizing noise rate, there exist quantum kernels with depths of $\Omega\left(1\right)$, $\Omega\left(\log\left(n\right)\right)$, and $\Omega\left(\operatorname{poly}\left(n\right)\right)$, respectively, where this expected difference vanishes.

**Strengths:**

While there have been previous studies on the sensitivity of quantum machine learning algorithms to the effects of noise, to-date so-called "variational" methods have been the focus of these studies. Here the authors give explicit bounds for quantum kernel methods, and give nicely define how their bound depends on relevant hyperparameters such as the depolarizing noise strength, the number of qubits of the model, the number of training samples, and so on. The authors also do nice numerical experiments to confirm their theoretical results also hold in practice.

**Weaknesses:**

The work is confusingly written, and certain concepts are not fully defined. For instance, "the kernel matrix" $K$ is used in Eq. (7) though is never given an explicit definition. There are also typos that need to be corrected, e.g., the title of Sec. 2.2 reads "qauntum" rather than "quantum." Furthermore, though Sec. 3 gives a concise rundown of the main Theorems proved in the paper, the implications are lost in the large algebraic expressions for the bounds---a quick explanation as to how these Theorems tie to Figure 1 would be extremely helpful in parsing the results. Finally, the overall picture is not too surprising---deep, noisy variational quantum machine learning algorithms also fail for similar reasons as the authors demonstrate quantum kernel methods fail.

**Questions:**

I think the main result is solid, and only recommend some structural changes in the paper for the main result to be clear. Currently Figure 1 is doing most of the heavy lifting in stating a clear result.

---

> ### Author Response · Authors · 2023-11-17
> **Response to Reviewer M7Gm**
>
> Thank you very much for your constructive comments and suggestions.
>
> Following your advice, we have added the needed definition and revised the typos. We have revised these paragraphs after Theorem 3.1 to better present the implications about the main result and to make it clearer how the main result ties to Figure 1. For more details, please see the revised paper (highlighted by a blue font for your convenience).
>
> We agree with you that the overall picture may not be too surprising. However, we explicitly depict the speed of the prediction concentration of noisy quantum kernels in terms of the strength of depolarization noise, the size of training samples, the number of qubits, the number of layers affected by quantum noise, and the number of measurement shots. This helps us quantitatively understand the limits of noisy quantum kernel methods. As an illustration, for a training sample set, whose scale is exponential in the number of qubits $N$, like $q^N$ for some $q>1$, noisy quantum kernel methods fail once the number of noisy layers exceeds $N\log\_{(1−\tilde{p} )^{−2}} q$.

---

> > ### Comment · Reviewer_M7Gm · 2023-11-21
> >
> > Thanks to the authors for adding much clarifying text; I've raised my score on the presentation accordingly.
> >
> > I unfortunately still agree with the other reviewers that---especially given the noise model of global depolarizing noise---the result is fairly straightforward. I understand the authors' argument that their noise model is "stronger" in the sense that a noise rate $p$ in the global model leaves the quantum state more intact than $n$ independent local depolarizing channels. While this is true at a surface level, both noise channels act differently---it is not just the case that the global channel is the local one at less noise---and the local model is typically thought of as being more realistic for experimental setups (as Reviewer VA13 mentioned, see arXiv:2210.11505 as an example of a paper analyzing the effects of local depolarizing noise rather than global depolarizing noise). Pushing the result in this direction would significantly strengthen the result.

---

> > > ### Author Response · Authors · 2023-11-22
> > >
> > > Thank you very much for your timely feedback. Really appreciated that you have raised your score on the presentation accordingly.
> > >
> > > We also appreciate your further advice about noise models. We completely agree with you that much work is worth being done to further investigate the effects of other local noises. We leave them for future research. Thank you for pointing out these potential research directions.
> > >
> > > Many thanks for your advice, suggestions and feedback, and wish you all the best.
> > >
> > > Best regards,
> > >
> > > Authors

---

> ### Author Response · Authors · 2023-11-22
> **Applicability to local depolarizing noise**
>
> Thank you so much for your insightful advice for investigating local depolarizing noise. We completely agree with you that "Pushing the result in this direction would significantly strengthen the result". We have worked diligently for a continuous span of ten days during the rebuttal period to explore this problem. Finally, we are excited to share with you that we find our result can be pushed to local depolarizing noise so that the prediction difference $\vert \tilde{h}-\bar{h} \vert$ converges to zero with the circuit depth increasing. Specifically, we outline the additional proof for local depolarizing noises as follows.
>
> Firstly, after 1-layer of $N$-fold local depolarizing channel, the relative entropy satisfies  $S(\mathcal{D}\_{\tilde{p}}^{\otimes N}(\rho\_{0})\Vert \frac{1}{D}I)\leq (1-\tilde{p})^2 S(\rho\_{0}\Vert \frac{1}{D}I)$ [1].
>
> Secondly, similar to Eq. (14) in our paper, denote by $\rho\_{\mathcal{D}}(x; x\_{i})$ the final state of the $2L$-layer local depolarizing noisy quantum circuit. Then, the corresponding kernel value is $\mathcal{K}(x,x\_{i})={\rm Tr} \left\[ P\_{0} \ \rho\_{\mathcal{D}}(x; x\_{i})\right\]$.
> For local-depolarizing noise, we have $\vert \mathcal{K}(x,x\_{i}) - \overline{\mathcal{K}}(x,x\_{i}) \vert \leq \Vert \rho\_{\mathcal{D}}(x; x\_{i})-\frac{1}{D}I \Vert\_{1} \leq  \sqrt{2 \ln 2\ S(\rho\_{\mathcal{D}}(x; x\_{i})\Vert\frac{1}{D}I)} \leq (1-\tilde{p})^{2L}\sqrt{2 \ln 2\ S(\rho\_{0}\Vert\frac{1}{D}I)}$, where the first two inequality come from the matrix H$\ddot{\rm o}$lder Inequality and quantum Pinsker Inequality [2], respectively. Thus, $\vert h(x)-\bar{h}(x) \vert = \mathcal{O}(n\vert \mathcal{K}(x,x\_{i}) - \overline{\mathcal{K}}(x,x\_{i}) \vert)=\mathcal{O}(n(1-\tilde{p})^{2L})$, where $n$ is the sample size.
>
> Thirdly, for our global depolarizing noise, since $S(\mathcal{N}\_{\tilde{p}} (\rho\_{0})\Vert \frac{1}{D}I)\leq (1-\tilde{p}) S(\rho\_{0}\Vert \frac{1}{D}I)$, we have $\vert h(x)-\bar{h}(x) \vert = \mathcal{O}(n(1-\tilde{p})^{L})$ by following a similar analysis.
>
> Combing the above statements, we can find that after local depolarizing noise, the kernel tends to get worse even for shallow circuits. The fundamental reason is that after an 1-layer of global depolarizing channel, the relative entropy decays as $S(\mathcal{N}\_{\tilde{p}} (\rho\_{0})\Vert \frac{1}{D}I)\leq (1-\tilde{p}) S(\rho\_{0}\Vert \frac{1}{D}I)$, while for the local depolarizing noise $S(\mathcal{D}\_{\tilde{p}}^{\otimes N}(\rho\_{0})\Vert \frac{1}{D}I)\leq (1-\tilde{p})^2 S(\rho\_{0}\Vert \frac{1}{D}I)$, which decays in a faster way.
>
> [1] Stilck França, D., García-Patrón, R. Limitations of optimization algorithms on noisy quantum devices. Nat. Phys. 17, 1221–1227 (2021). https://doi.org/10.1038/s41567-021-01356-3
>
> [2] Watrous, J. The Theory of Quantum Information. Cambridge: Cambridge University Press. https://doi.org/10.1017/9781316848142, Theorem 5.38 for quantum Pinsker Inequality.

---

### Official Review · Reviewer_5XBn · 2023-10-29

**Soundness:** 3 good
**Presentation:** 3 good
**Contribution:** 2 fair
**Rating:** 5
**Confidence:** 4

**Summary:**

Quantum machine learning (QML) is an emerging field that explores the power of quantum models for machine learning tasks. Quantum kernel methods have shown promise in various applications. The paper focuses on the prediction capability and limitations of quantum kernel methods under quantum depolarization noise in the NISQ era. It aims to understand the impact of noise on the performance of quantum kernel methods. The paper also presents theoretical bounds and extends the analysis to quantum circuits with certain width and depth.

**Strengths:**

One strength of this paper is its focus on the impact of noise on quantum kernel methods. The authors provide insights into the behavior of quantum kernels when exposed to noise, specifically global depolarization noise. They theoretically characterize the concentration speed of predictions of the optimal hypothesis inferred by noisy quantum kernels in terms of several basic factors. This result is meaningful to understand the limitation of quantum kernel model in the NISQ era.

**Weaknesses:**

- The noise model is too ideal and limited. It is just a global quantum depolarization noise. For guidelines on characterizing the power of quantum kernel models on noisy devices, it is important to consider more realistic noises. Such strong assumptions as global depolarization noise may not accurately represent real-world noise scenarios.
- There are several previous works on noisy quantum kernels, such as Wang et al. (2021), Stilck Franca & Garcia-Patron (2021); De Palma et al. (2023), Thanasilp et al. (2022). It is not clear how this work compares to existing research and what new limitations are discovered in this paper. The advantages or significance of this work is not clear enough to me. It would be better if the authors could explain this angle in a more organized way.

**Questions:**

- What are the key new findings on the noisy quantum kernel model compared with previous works?
- Why does the paper only consider the global depolarizing noise? Is this for theoretical convenience?

**Details Of Ethics Concerns:**

NA.

---

> ### Author Response · Authors · 2023-11-17
> **Response to Reviewer 5XBn**
>
> Thank you very much for your helpful comments and suggestions.
>
> First of all, we would like to point out that, for the problem of power characterization under consideration, our global noise model is actually weaker than the noise models considered in previous works you mentioned, such as, Stilck Franca & Garcia-Patron (2021), De Palma et al. (2023) and Thanasilp et al. (2022).
>
> Specifically, in our work, the noise model, $\mathcal{N}\_{\tilde{p}}=(1-\tilde{p})\rho+\tilde{p}\frac{1}{D}I$ with rate $\tilde{p}$, is referred to as “global” as it applies onto all qubits after each layer of ideal unitary gates. While in Stilck Franca & Garcia-Patron (2021) and De Palma et al. (2023), the considered noise model is the local depolarizing noise. However, it applies to each qubit. Thus, their noise channel applied after each ideal unitary layer is $\mathcal{D}\_{\tilde{p}}^{\otimes N}$, with $\mathcal{D}\_{\tilde{p}}=(1-\tilde{p})\rho+\frac{1}{2}I$ and $N$ being the number of qubits. We have rewritten it using our notation for convenience of comparison. With the same noise rate $\tilde{p}$, we can compare our global depolarizing noise $\mathcal{N}\_{\tilde{p}}$ with its counterpart $\mathcal{D}\_{\tilde{p}}^{\otimes N}$. It can be seen that under our global noise, the quantum state is left untouched with probability $1-\tilde{p}$, while under the so-called local noise, the probability of the state remaining unchanged is only $(1-\tilde{p})^N$, which is exponentially small in $N$. Thus, in this sense our global depolarizing model is weaker than the so-called local depolarizing model. As for the Pauli noise model considered in Thanasilp et al. (2022), following a similar analysis, we can see that the Pauli noise is also stronger than ours.
>
> When presenting negative results concerning noisy quantum kernels, it is better to assume a relatively weaker noise model. Once the kernel methods fail under weaker noises, they fail under stronger ones in general. Note that the noise rate $\tilde{p}>0$ in our work can be arbitrarily small. Thus, it can depict the case where the noise influence is very weak, namely, after the noise the quantum state is left untouched with a very high probability $1-\tilde{p}$. Thus, in this sense, our global depolarizing model is an appropriate starting point to study the limitations of noisy quantum kernel methods. We have made this clearer in the revised manuscript after Eq. (12). As you suggested, much work needs to be done to further investigate the limitations of quantum kernel methods, e.g., under more realistic quantum noise. We leave this for future research.
>
> To address your constructive question "What are the key new findings on the noisy quantum kernel model compared with previous works", we have thoroughly revised Sec. 1.2 Related Work to compare our work with previous works. Your question definitely helps us to present our key new findings and the differences from the existing works more clearly. For more details, please refer to revised Sec. 1.2 Related Work as well as the above response to your question regarding global depolarizing noise.

---

> ### Author Response · Authors · 2023-11-22
>
> Dear Reviewer 5XBn,
>
> Many thanks for your dedicated time to review our work. We have taken your insightful comments seriously and have made necessary modifications based on your suggestions. With the imminent closure of the discussion period, time is of the essence.
>
> Given that our work received your’s less favorable scores, we worked diligently to clarify our assumption and difference from prior work during the rebuttal period, demonstrating our unwavering commitment to addressing every facet of your feedback.
>
> We kindly request your esteemed attention to our rebuttal response. Your responsible and positive endorsement holds paramount significance during this critical phase.
>
> Wishing you all the best and awaiting your valued evaluation.
>
> Best regards,
>
> Authors

---

> ### Author Response · Authors · 2023-11-22
> **Extension to other noises**
>
> Thank you so much for your insightful advice for investigating other noise. We have worked diligently for a continuous span of ten days during the rebuttal period to explore this problem. Finally, we are excited to share with you that we find our result can be pushed to local depolarizing noise so that the prediction difference $\vert \tilde{h}-\bar{h} \vert$ converges to zero with the circuit depth increasing. Specifically, we outline the additional proof for local depolarizing noises as follows.
>
> Firstly, after 1-layer of $N$-fold local depolarizing channel, the relative entropy satisfies  $S(\mathcal{D}\_{\tilde{p}}^{\otimes N}(\rho\_{0})\Vert \frac{1}{D}I)\leq (1-\tilde{p})^2 S(\rho\_{0}\Vert \frac{1}{D}I)$ [1].
>
> Secondly, similar to Eq. (14) in our paper, denote by $\rho\_{\mathcal{D}}(x; x\_{i})$ the final state of the $2L$-layer local depolarizing noisy quantum circuit. Then, the corresponding kernel value is $\mathcal{K}(x,x\_{i})={\rm Tr} \left\[ P\_{0} \ \rho\_{\mathcal{D}}(x; x\_{i})\right\]$.
> For local-depolarizing noise, we have $\vert \mathcal{K}(x,x\_{i}) - \overline{\mathcal{K}}(x,x\_{i}) \vert \leq \Vert \rho\_{\mathcal{D}}(x; x\_{i})-\frac{1}{D}I \Vert\_{1} \leq  \sqrt{2 \ln 2\ S(\rho\_{\mathcal{D}}(x; x\_{i})\Vert\frac{1}{D}I)} \leq (1-\tilde{p})^{2L}\sqrt{2 \ln 2\ S(\rho\_{0}\Vert\frac{1}{D}I)}$, where the first two inequality come from the matrix H$\ddot{\rm o}$lder Inequality and quantum Pinsker Inequality [2], respectively. Thus, $\vert h(x)-\bar{h}(x) \vert = \mathcal{O}(n\vert \mathcal{K}(x,x\_{i}) - \overline{\mathcal{K}}(x,x\_{i}) \vert)=\mathcal{O}(n(1-\tilde{p})^{2L})$, where $n$ is the sample size.
>
> Thirdly, for our global depolarizing noise, since $S(\mathcal{N}\_{\tilde{p}} (\rho\_{0})\Vert \frac{1}{D}I)\leq (1-\tilde{p}) S(\rho\_{0}\Vert \frac{1}{D}I)$, we have $\vert h(x)-\bar{h}(x) \vert = \mathcal{O}(n(1-\tilde{p})^{L})$ by following a similar analysis.
>
> Combing the above statements, we can find that after local depolarizing noise, the kernel tends to get worse even for shallow circuits. The fundamental reason is that after an 1-layer of global depolarizing channel, the relative entropy decays as $S(\mathcal{N}\_{\tilde{p}} (\rho\_{0})\Vert \frac{1}{D}I)\leq (1-\tilde{p}) S(\rho\_{0}\Vert \frac{1}{D}I)$, while for the local depolarizing noise $S(\mathcal{D}\_{\tilde{p}}^{\otimes N}(\rho\_{0})\Vert \frac{1}{D}I)\leq (1-\tilde{p})^2 S(\rho\_{0}\Vert \frac{1}{D}I)$, which decays in a faster way.
>
> [1] Stilck França, D., García-Patrón, R. Limitations of optimization algorithms on noisy quantum devices. Nat. Phys. 17, 1221–1227 (2021). https://doi.org/10.1038/s41567-021-01356-3
>
> [2] Watrous, J. The Theory of Quantum Information. Cambridge: Cambridge University Press. https://doi.org/10.1017/9781316848142, Theorem 5.38 for quantum Pinsker Inequality.

---

> > ### Comment · Reviewer_5XBn · 2023-11-23
> >
> > Thanks for the replies and arguments. However, it is still unclear if the main results on global depolarizing noise apply to the quantum kernel model under more usual and meaningful noise models such as local depolarizing noise. This clarification is necessary since the global noise is less interesting, and different types of noise can have varied effects on quantum kernel methods. Hence I will keep my score unchanged.

---

> > > ### Author Response · Authors · 2023-11-23
> > >
> > > Thank you for your feedback. We have proven that the main results also apply to the quantum kernel model under more usual and meaningful noise models such as local depolarizing noise. Here are the main proof steps.
> > >
> > > Firstly, after 1-layer of $N$-fold local depolarizing channel, the relative entropy satisfies  $S(\mathcal{D}\_{\tilde{p}}^{\otimes N}(\rho\_{0})\Vert \frac{1}{D}I)\leq (1-\tilde{p})^2 S(\rho\_{0}\Vert \frac{1}{D}I)$ [1].
> > >
> > > Secondly, similar to Eq. (14) in our paper, denote by $\rho\_{\mathcal{D}}(x; x\_{i})$ the final state of the $2L$-layer local depolarizing noisy quantum circuit. Then, the corresponding kernel value is $\mathcal{K}(x,x\_{i})={\rm Tr} \left\[ P\_{0} \ \rho\_{\mathcal{D}}(x; x\_{i})\right\]$.
> > > For local-depolarizing noise, we have $\vert \mathcal{K}(x,x\_{i}) - \overline{\mathcal{K}}(x,x\_{i}) \vert \leq \Vert \rho\_{\mathcal{D}}(x; x\_{i})-\frac{1}{D}I \Vert\_{1} \leq  \sqrt{2 \ln 2\ S(\rho\_{\mathcal{D}}(x; x\_{i})\Vert\frac{1}{D}I)} \leq (1-\tilde{p})^{2L}\sqrt{2 \ln 2\ S(\rho\_{0}\Vert\frac{1}{D}I)}$, where the first two inequality come from the matrix H$\ddot{\rm o}$lder Inequality and quantum Pinsker Inequality [2], respectively. Thus, $\vert h(x)-\bar{h}(x) \vert = \mathcal{O}(n\vert \mathcal{K}(x,x\_{i}) - \overline{\mathcal{K}}(x,x\_{i}) \vert)=\mathcal{O}(n(1-\tilde{p})^{2L})$, where $n$ is the sample size.
> > >
> > > Thirdly, for our global depolarizing noise, since $S(\mathcal{N}\_{\tilde{p}} (\rho\_{0})\Vert \frac{1}{D}I)\leq (1-\tilde{p}) S(\rho\_{0}\Vert \frac{1}{D}I)$, we have $\vert h(x)-\bar{h}(x) \vert = \mathcal{O}(n(1-\tilde{p})^{L})$ by following a similar analysis.
> > >
> > > Combing the above statements, we can find that after local depolarizing noise, the kernel tends to get worse even for shallow circuits. The fundamental reason is that after an 1-layer of global depolarizing channel, the relative entropy decays as $S(\mathcal{N}\_{\tilde{p}} (\rho\_{0})\Vert \frac{1}{D}I)\leq (1-\tilde{p}) S(\rho\_{0}\Vert \frac{1}{D}I)$, while for the local depolarizing noise $S(\mathcal{D}\_{\tilde{p}}^{\otimes N}(\rho\_{0})\Vert \frac{1}{D}I)\leq (1-\tilde{p})^2 S(\rho\_{0}\Vert \frac{1}{D}I)$, which decays in a faster way.
> > >
> > > [1] Stilck França, D., García-Patrón, R. Limitations of optimization algorithms on noisy quantum devices. Nat. Phys. 17, 1221–1227 (2021). https://doi.org/10.1038/s41567-021-01356-3
> > >
> > > [2] Watrous, J. The Theory of Quantum Information. Cambridge: Cambridge University Press. https://doi.org/10.1017/9781316848142, Theorem 5.38 for quantum Pinsker Inequality.
> > >
> > > We do expect that our further clarification addresses your concerns. Nevertheless, the final decision for endorsement or rejection ultimately lies with you, and we deeply respect that.
> > >
> > > Many thanks for your advice, suggestions and feedback, and wish you all the best.
> > >
> > > Best regards,
> > >
> > > Authors

---

### Official Review · Reviewer_VA13 · 2023-10-29

**Soundness:** 2 fair
**Presentation:** 4 excellent
**Contribution:** 2 fair
**Rating:** 5
**Confidence:** 4

**Summary:**

The paper studies the question of how having global depolarization noise in the quantum device impacts the prediction performance of a quantum kernel method. The paper provides comprehensive analytical results bounding the generalization errors and conduct numerical experiments to illustrate the analytical results.

**Strengths:**

The problem of noise in quantum devices is a very significant one. Studying the impact of noise on quantum machine learning models is an important question.

The authors provide a detailed and comprehensive theoretical analysis of the impact of global depolarizing noise on quantum kernel methods (one of the most promising models for quantum machine learning).

**Weaknesses:**

The paper focuses on global depolarizing noise, which is not considered realistic for quantum devices (as well as noisy quantum kernel methods). The natural noise model would be local depolarizing noise, where each qubit is subject to a small amount of noise.

Global depolarizing noise has a simple algebraic structure. Hence, the generalization error bounds provided in this work are relatively straightforward to derive.

The key behaviors of noisy quantum kernel methods illustrated by the analytical results are expected. As the total noise increases, the generalization error decreases while the training error increases. Hence, the generalization error can be close to zero while the prediction performance is bad.

**Questions:**

- The authors should provide analytical results based on local depolarization noise. Techniques from https://arxiv.org/abs/2210.11505 and related works should be helpful in this case.

- The numerical experiments can be improved to showcase the dependence on other parameters such as system size and noise rate.

---

> ### Author Response · Authors · 2023-11-17
> **Response to Reviewer VA13**
>
> Thank you very much for your helpful comments and suggestions as well as pointing out the useful reference.
>
> - The authors should provide analytical results based on local depolarization noise. Techniques from https://arxiv.org/abs/2210.11505 and related works should be helpful in this case.
>
> We agree with you that the local depolarizing noise is a natural noise model, where each qubit is subject to a small amount of noise. As compared with it, our global depolarizing model is a stronger assumption about the noise at first glance. However, on second thought, our global depolarizing model is actually weaker than the so-called local one for our problem in this paper. The reason is as follows.
>
> In our work, as illustrated in Fig. 2(c), after each layer of ideal unitary gates, a depolarizing channel, $\mathcal{N}\_{\tilde{p}}=(1-\tilde{p})\rho+\tilde{p}\frac{1}{D}I$ with rate $\tilde{p}$ is applied. We refer to it “global” as it applies onto all qubits. In your mentioned paper ( https://arxiv.org/abs/2210.11505), two kinds of noise models are considered. The first one is the so-called local depolarizing noise acting on each qubit, which reads $\mathcal{D}\_{\tilde{p}}=(1-\tilde{p})\rho+\frac{1}{2}I$. We have rewritten it using our notation for convenience of comparison. As demonstrated by Fig. 1 in the mentioned paper, the noise channel applied after each layer of ideal unitary gates should be $\mathcal{D}\_{\tilde{p}}^{\otimes N}$ with $N$ being the number of qubits. Now we can compare our global depolarizing noise $\mathcal{N}\_{\tilde{p}}$ with its counterpart $\mathcal{D}\_{\tilde{p}}^{\otimes N}$ under the same noise rate $\tilde{p}$. It can be seen that under our global depolarizing noise, the quantum state is left untouched with probability $1-\tilde{p}$, while under the so-called local noise, the probability of the state remaining unchanged is only $(1-\tilde{p})^N$, which is exponentially small in $N$. Therefore, in this sense our global depolarizing model is weaker than the so-called local depolarizing model. Another considered noise model in the mentioned paper is non-unital noise acting on each qubit. Following a similar analysis to the above local depolarizing noise, it can be seen that the non-unital noise is also stronger than ours.
>
> When presenting negative results concerning noisy quantum kernels, it is better to assume a relatively weaker noise model. Once the kernel methods fail under weaker noises, they fail under stronger ones in general. Note that the noise rate $\tilde{p}>0$ in our work can be arbitrarily small. Thus, it can depict the case where the noise influence is very weak, namely, after the noise the quantum state is left untouched with a very high probability $1-\tilde{p}$. Thus, in this sense, our global depolarizing model is an appropriate starting point to study the limitations of noisy quantum kernel methods. We have made this clearer in the revised manuscript after Eq. (12). We completely agree with you that much work needs to be done to further investigate the limitations of quantum kernel methods under more realistic quantum noise. We leave this for our future research.
>
> We agree with you that the key behaviors of noisy quantum kernel methods are expected. However, one of our key contributions is that we explicitly depict the speed of the prediction concentration of noisy quantum kernels in terms of the strength of depolarization noise, the size of training samples, the number of qubits, the number of layers affected by quantum noise, and the number of measurement shots. This helps us quantitatively understand the limits of noisy quantum kernel methods. As an illustration, for a training sample set, whose scale is exponential in the number of qubits $N$, like $q^N$ for some $q>1$, noisy quantum kernel methods fail once the number of noisy layers exceeds $N\log\_{(1−\tilde{p} )^{−2}} q$.
>
> - The numerical experiments can be improved to showcase the dependence on other parameters such as system size and noise rate.
>
> We agree with you that the numerical results can be improved to showcase the dependence on other parameters such as system size and noise rate. In our paper, since we mainly focus on the influence of noise accumulation on the capability of quantum kernel methods, we only illustrate the dependence of the hypothesis function and the training error as well as the test error on the number of noisy layers. There are many open problems for future research.

---

> > ### Comment · Reviewer_VA13 · 2023-11-21
> >
> > I do not think the argument presented by the authors convinces me regarding the suitability of global depolarizing noise. When considering the limitations of a method, it is true that a model that fails for a weaker noise model will fail for a stronger noise model. But this is not the case here. The authors try to argue that global depolarizing noise is "weaker than" local depolarizing noise, but the argument is simply incomplete. For example, it is impossible to perform quantum error correction when the noise is a global depolarizing noise. However, one can perform quantum error correction for local depolarizing noise. Global depolarizing noise and local depolarizing noise are incomparable noise models with different characteristics. Hence, it is not clear if the main results in this work focusing on global depolarizing noise apply to local depolarizing noise.
> >
> > The authors did not include new numerical experiments showcasing the dependence on other parameters.
> >
> > Overall, I do not think the authors addressed any of my questions and comments. Hence, my score remains unchanged.

---

> ### Author Response · Authors · 2023-11-22
> **Applicability to local depolarizing noise**
>
> Thank you so much for your constructive advice for investigating local depolarizing noise. We have worked diligently for a continuous span of ten days during the rebuttal period to explore this problem. Finally, we are excited to share with you that we find our main result can be pushed to local depolarizing noise so that the prediction difference $\vert \tilde{h}-\bar{h} \vert$ converges to zero with the circuit depth increasing. Specifically, we outline the additional proof for local depolarizing noises as follows.
>
> Firstly, after 1-layer of $N$-fold local depolarizing channel, the relative entropy satisfies  $S(\mathcal{D}\_{\tilde{p}}^{\otimes N}(\rho\_{0})\Vert \frac{1}{D}I)\leq (1-\tilde{p})^2 S(\rho\_{0}\Vert \frac{1}{D}I)$ [1].
>
> Secondly, similar to Eq. (14) in our paper, denote by $\rho\_{\mathcal{D}}(x; x\_{i})$ the final state of the $2L$-layer local depolarizing noisy quantum circuit. Then, the corresponding kernel value is $\mathcal{K}(x,x\_{i})={\rm Tr} \left\[ P\_{0} \ \rho\_{\mathcal{D}}(x; x\_{i})\right\]$.
> For local-depolarizing noise, we have $\vert \mathcal{K}(x,x\_{i}) - \overline{\mathcal{K}}(x,x\_{i}) \vert \leq \Vert \rho\_{\mathcal{D}}(x; x\_{i})-\frac{1}{D}I \Vert\_{1} \leq  \sqrt{2 \ln 2\ S(\rho\_{\mathcal{D}}(x; x\_{i})\Vert\frac{1}{D}I)} \leq (1-\tilde{p})^{2L}\sqrt{2 \ln 2\ S(\rho\_{0}\Vert\frac{1}{D}I)}$, where the first two inequality come from the matrix H$\ddot{\rm o}$lder Inequality and quantum Pinsker Inequality [2], respectively. Thus, $\vert h(x)-\bar{h}(x) \vert = \mathcal{O}(n\vert \mathcal{K}(x,x\_{i}) - \overline{\mathcal{K}}(x,x\_{i}) \vert)=\mathcal{O}(n(1-\tilde{p})^{2L})$, where $n$ is the sample size.
>
> Thirdly, for our global depolarizing noise, since $S(\mathcal{N}\_{\tilde{p}} (\rho\_{0})\Vert \frac{1}{D}I)\leq (1-\tilde{p}) S(\rho\_{0}\Vert \frac{1}{D}I)$, we have $\vert h(x)-\bar{h}(x) \vert = \mathcal{O}(n(1-\tilde{p})^{L})$ following a similar analysis.
> Combing the above statements, we can find that after local depolarizing noise, the kernel tends to get worse even for shallow circuits. The fundamental reason is that after an 1-layer of global depolarizing channel, the relative entropy decays as $S(\mathcal{N}\_{\tilde{p}} (\rho\_{0})\Vert \frac{1}{D}I)\leq (1-\tilde{p}) S(\rho\_{0}\Vert \frac{1}{D}I)$, while for the local depolarizing noise $S(\mathcal{D}\_{\tilde{p}}^{\otimes N}(\rho\_{0})\Vert \frac{1}{D}I)\leq (1-\tilde{p})^2 S(\rho\_{0}\Vert \frac{1}{D}I)$, which decays in a faster way.
>
> In the current quantum devices, the noise is vital of consideration before realizing quantum error correction and fault tolerant quantum computing. To this end, we and the authors in many Related Works are focusing on the effect of quantum noise and provide a crucial warning to utilize noisy quantum kernels under different noise models.
>
> In a word, for the ultimate goal of machine learning, i.e., prediction, the local depolarizing noise model is usually “stronger” than our global one. Our work is especially useful for developing quantum kernel methods in noisy intermediate-scale quantum (NISQ) era where we have  insufficient capability for quantum error correction. We agree with you that further work is worthing being presented to discuss the different performance and effects of quantum error correction and quantum error mitigation for global depolarizing noise and local depolarizing noise.
>
> [1] Stilck França, D., García-Patrón, R. Limitations of optimization algorithms on noisy quantum devices. Nat. Phys. 17, 1221–1227 (2021). https://doi.org/10.1038/s41567-021-01356-3
>
> [2] Watrous, J. The Theory of Quantum Information. Cambridge: Cambridge University Press. https://doi.org/10.1017/9781316848142, Theorem 5.38 for quantum Pinsker Inequality.
>
> We apologize that we did not include new numerical experiments in the revised paper due to limited time. We had focused on presenting more convincing explanations on the employment of global depolarizing noise and the critical applicability issue on local depolarizing noise. We have started to implement experiments to obtain additional numerical results but it will take too much time to generate sufficient results showcasing the dependence on other parameters (e.g., a simple experiment needs to run more than 30 hours). In principle, we do not find any technical difficulty at this stage and will present additional results in the near future. We appreciate your constructive feedback and advice.

---

### Official Review · Reviewer_8JA5 · 2023-11-09

**Soundness:** 2 fair
**Presentation:** 3 good
**Contribution:** 2 fair
**Rating:** 3
**Confidence:** 3

**Summary:**

This paper studies the effect of noise on the performance of quantum kernel methods.  Quantum kernel methods can perform kernel computation on a quantum computer potentially faster than classical computers. This paper studies the effect of NISQ noise and infidelity on kernel methods. Particularly, they study the depolarisation channel (as a noise model) on the power of quantum kernel computations.

**Strengths:**

The main strength of this paper is a series of negative theoretical studies on the effect of depolarisation noise. This is an important topic, especially in near-term quantum computers. The paper looks solid, although I did not check all the proofs.

**Weaknesses:**

My major issue with the paper is the correctness of the main result (Theorem 3.1). I might be missing something, as I am not convinced this theorem is correct. Equation (20) in the theorem bounds the distance between the quantum kernel's choice $\tilde{h}$ and the worst possible predictor $\bar{h}$. The bound is in terms of the noise bias $p$, the number of samples (n), and the dimension (D).
Based on this bound $|\bar{h} -\tilde{h}|$ converges to zero as $n$ and $D$ grow large regardless of the value of p<1!

Surprisingly, if we set $p=0$, implying no noise, the bound converges to zero! This does not make any sense as we expect when $p=0$, the kernel method should work better than the worst predictor! Am I missing something?

**Questions:**

Q1. Why does the bound in Theorem 3.1 still converge to zero as $p=0$?
Q2. The samples are classical in this work, but the kernel is quantum. What can be done with your work for the quantum samples?

---

> ### Author Response · Authors · 2023-11-17
> **Response to Reviewer 8JA5**
>
> Thank you very much for your insightful comments.
>
> -  Why does the bound in Theorem 3.1 still converge to zero as $p=0$?
>
> As you mentioned, Equation (20) in the theorem bounds the distance between the quantum kernel's choice $\tilde{h}$ and the worst possible predictor $\bar{h}$. It consists of three terms. The third term converges to 0 as $n$ increases, and the second term approaches to 0 as $n$ and $D$ increase (as long as $n$ is not in the order of $D$). While the first term $f(\cdot)$ goes to 0 if and only if its argument goes to 0. If we set $p=0$, then the argument of the first term $f(\cdot)$ is a constant of $\frac{n}{\lambda}(1+\frac{1}{D})$, and in turn the bound does not converge to zero. Thank you for pointing out this potential ambiguity. Following your insightful comments, we have made it clearer in the revised manuscript.
>
> - The samples are classical in this work, but the kernel is quantum. What can be done with your work for the quantum samples?
>
> In our work, the samples are classical and encoded into quantum states to compute their kernels. If the samples are quantum, our technique can still be applied to analyze the impact of noise on quantum kernel methods if the depolarizing noise occurs during the generation of quantum samples or during the computation of kernels. We leave this as future work for a more detailed analysis. Thank you for your insightful comments.

---

> > ### Comment · Reviewer_8JA5 · 2023-11-23
> > **Response to the authors**
> >
> > Thank you for your detailed response. I think I made a typo in my review; I meant $\tilde{p}=0$ so that $(1-p)=0$ and the argument of $f(\cdot)$ becomes zero. In that case, the distance between the quantum kernel's choice and the worst possible predictor converges to zero. But this is surprising, as $\tilde{p}=0$ means there is no noise!

---

> ### Author Response · Authors · 2023-11-22
>
> Dear Reviewer 8JA5,
>
> Many thanks for your dedicated time to review our work. We have taken your comments seriously and have made the necessary explanations based on your suggestions. With the imminent closure of the discussion period, time is of the essence.
>
> Given that our work received your’s less favorable scores, we worked diligently to clarify the technical details during the rebuttal period, demonstrating our unwavering commitment to addressing every facet of your feedback.
>
> We kindly request your esteemed attention to our rebuttal response. Your responsible and positive endorsement holds paramount significance during this critical phase.
>
> Wishing you all the best and awaiting your valued evaluation.
>
> Best regards,
>
> Authors

---

> ### Author Response · Authors · 2023-11-23
> **New results on applicability to local depolarizing noises**
>
> Dear Reviewer 8JA5,
>
> Thank you so much for reviewing our paper. We have worked diligently for a continuous span of ten days during the rebuttal period to explore this issue on local depolarizing noise. Finally, we are excited to share with you that we find our main result can be pushed to local depolarizing noise so that the prediction difference $\vert \tilde{h}-\bar{h} \vert$ converges to zero with the circuit depth increasing. Specifically, we outline the additional proof for local depolarizing noises as follows.
>
> Firstly, after 1-layer of $N$-fold local depolarizing channel, the relative entropy satisfies  $S(\mathcal{D}\_{\tilde{p}}^{\otimes N}(\rho\_{0})\Vert \frac{1}{D}I)\leq (1-\tilde{p})^2 S(\rho\_{0}\Vert \frac{1}{D}I)$ [1].
>
> Secondly, similar to Eq. (14) in our paper, denote by $\rho\_{\mathcal{D}}(x; x\_{i})$ the final state of the $2L$-layer local depolarizing noisy quantum circuit. Then, the corresponding kernel value is $\mathcal{K}(x,x\_{i})={\rm Tr} \left\[ P\_{0} \ \rho\_{\mathcal{D}}(x; x\_{i})\right\]$.
> For local-depolarizing noise, we have $\vert \mathcal{K}(x,x\_{i}) - \overline{\mathcal{K}}(x,x\_{i}) \vert \leq \Vert \rho\_{\mathcal{D}}(x; x\_{i})-\frac{1}{D}I \Vert\_{1} \leq  \sqrt{2 \ln 2\ S(\rho\_{\mathcal{D}}(x; x\_{i})\Vert\frac{1}{D}I)} \leq (1-\tilde{p})^{2L}\sqrt{2 \ln 2\ S(\rho\_{0}\Vert\frac{1}{D}I)}$, where the first two inequality come from the matrix H$\ddot{\rm o}$lder Inequality and quantum Pinsker Inequality [2], respectively. Thus, $\vert h(x)-\bar{h}(x) \vert = \mathcal{O}(n\vert \mathcal{K}(x,x\_{i}) - \overline{\mathcal{K}}(x,x\_{i}) \vert)=\mathcal{O}(n(1-\tilde{p})^{2L})$, where $n$ is the sample size.
>
> Thirdly, for our global depolarizing noise, since $S(\mathcal{N}\_{\tilde{p}} (\rho\_{0})\Vert \frac{1}{D}I)\leq (1-\tilde{p}) S(\rho\_{0}\Vert \frac{1}{D}I)$, we have $\vert h(x)-\bar{h}(x) \vert = \mathcal{O}(n(1-\tilde{p})^{L})$ by following a similar analysis.
>
> Combing the above statements, we can find that after local depolarizing noise, the kernel tends to get worse even for shallow circuits. The fundamental reason is that after an 1-layer of global depolarizing channel, the relative entropy decays as $S(\mathcal{N}\_{\tilde{p}} (\rho\_{0})\Vert \frac{1}{D}I)\leq (1-\tilde{p}) S(\rho\_{0}\Vert \frac{1}{D}I)$, while for the local depolarizing noise $S(\mathcal{D}\_{\tilde{p}}^{\otimes N}(\rho\_{0})\Vert \frac{1}{D}I)\leq (1-\tilde{p})^2 S(\rho\_{0}\Vert \frac{1}{D}I)$, which decays in a faster way.
>
> In the current quantum devices, the noise is vital of consideration before realizing quantum error correction and fault tolerant quantum computing. To this end, we and the authors in many Related Works are focusing on the effect of quantum noise and provide a crucial warning to utilize noisy quantum kernels under different noise models.
>
> In a word, for the ultimate goal of machine learning, i.e., prediction, the local depolarizing noise model is usually “stronger” than our global one. Our work is especially useful for developing quantum kernel methods in noisy intermediate-scale quantum (NISQ) era where we have no sufficient capability for quantum error correction.
>
> [1] Stilck França, D., García-Patrón, R. Limitations of optimization algorithms on noisy quantum devices. Nat. Phys. 17, 1221–1227 (2021). https://doi.org/10.1038/s41567-021-01356-3
>
> [2] Watrous, J. The Theory of Quantum Information. Cambridge: Cambridge University Press. https://doi.org/10.1017/9781316848142, Theorem 5.38 for quantum Pinsker Inequality.
>
> We do expect that our further clarification addresses your concerns. Nevertheless, the final decision for endorsement or rejection ultimately lies with you, and we deeply respect that.
>
> Many thanks for your advice, suggestions and feedback, and wish you all the best.
>
> Best regards,
>
> Authors

---

> ### Author Response · Authors · 2023-11-23
> **Response to Reviewer 8JA5**
>
> Thank you very much for your feedback.
>
> In fact, our main result holds only for the noisy case. Note that the function $f(z)$ in the RHS of Eq. (20) has the form of $f\left( z\right) =\frac{z+8\sqrt{\frac{z}{\lambda}}}{1-z}$. To make it meaningful, its argument must be less than 1. Thus, in Eq. (20), $\frac{n}{\lambda}(1-p)(1+\frac{1}{D}) $ should be less than 1. This corresponds to noisy channels rather than the ideal case. This is because $1-p=(1-\tilde{p})^{2L}$. In the idea case $\tilde{p}=0$, then $1-p=1$. This leads to $\frac{n}{\lambda}(1-p)(1+\frac{1}{D}) > 1$.
>
> To sum up, in this paper, we focus on the effect of quantum noise and provide a crucial warning to utilize noisy quantum kernels. Our main results are indicative for noisy deep quantum circuits.

---

### Author Response · Authors · 2023-11-20
**General response**

Dear reviewers and meta reviewers,

We appreciate your valuable comments and constructive suggestions, which really help improve the quality of this paper. We have thoroughly revised the paper with changes highlighted in a BLUE font. For your convenience, we briefly list the main changes as follows.

Firstly, we have further clarified why we mainly consider global depolarizing noise and justified why the noise assumption is weaker than other noise models for the problem under consideration.

Secondly, we further highlight the difference of our work from prior works and present our contributions more clearly.

Lastly, we have also addressed all technical questions in detail.

We kindly request you to inform us if there are any other concerns that might have been overlooked, or if you have new questions. Please feel free to put your further comments on OpenReview and we are always happy to answer any questions from you.

---

### Author Response · Authors · 2023-11-22
**Extension to local depolarizing noise**

Thank you so much for your constructive comments. We have worked diligently for a continuous span of ten days during the rebuttal period to explore this issue on local depolarizing noise. Finally, we are excited to share with you that we find our main result can be pushed to local depolarizing noise. Specifically, we outline the proof for local depolarizing noises in the response to Reviewers.

---

### Meta-Review · Area_Chair_LnWf · 2023-12-03

**Metareview:**

The paper investigates the effects of noise on quantum kernel methods in machine learning. Their theoretical result characterizes the expected difference of the predictions between the optimal hypothesis under the depolarization noise and the worst hypothesis, highlighting the role of the noise, number of training samples, and number of qubits.

A common concern among the reviewers is that the noise model is too limited. Although the authors tried to address this point during the discussion period, the reviewers are in general not convinced by the arguments made regarding the suitability of the depolorizing noise assumptions made by the paper.
Another concern was the lack of discussion of some prior works. While the authors also tried to address this point, this has led to significant changes, which would require another round of reviews.

Overall, the paper is unfortunately not yet ready for acceptance. I strongly encourage the authors to address the comments from the reviewers, i.e. expand the related work discussion and the discussion of the theoretical and empirical results, as well as also expand the discussion of the assumptions made by the noise model employed in the paper.

**Justification For Why Not Higher Score:**

The paper is clearly below the acceptance threshold. The authors made a significant number of changes during the discussion period, which I think requires a fresh evaluation of the paper.

**Justification For Why Not Lower Score:**

N/A

---

### Decision · Program_Chairs · 2024-01-16

Reject